# Context-dependent serotonin signaling links dietary quality to foraging decisions

Likui Feng [1], Javier Marquina-Solis[1,2], Lishu Yue [1], Audrey Harnagel[1,3], Yarden Greenfeld [1] & Cornelia I. Bargmann [1] ✉

Animals sense their metabolic needs to guide adaptive behaviors partly through serotonin, a neurotransmitter associated with feeding in many species. Here we investigate the ability of the serotonin system to evaluate and interpret diverse diets by studying long-term foraging behaviors of the nematode *C. elegans* on bacteria. Behavioral screens on a genome-wide collection of *E coli* strains identified 22 metabolic mutants that induce behavioral aversion and stress responses in *C. elegans*. We show that different classes of serotonergic neurons promote aversion to non-preferred *E. coli* diets and retention on preferred *E. coli* diets, respectively, through different serotonin receptors. Serotonin is integrated with dopamine and octopamine signals across distributed circuits to direct opposing behavioral responses to preferred and aversive diets. These results reveal interacting neuromodulatory circuits that guide context-dependent evaluation of dietary quality.

Animals discriminate between high-quality and low-quality food to optimize their development, reproduction, and physiology. While food quality can be sensed through olfactory and gustatory cues, internally-sensed metabolic needs also drive food preferences. For example, an interoceptive dopaminergic circuit allows *Drosophila* to reject an incomplete diet lacking essential amino acids[1]. Preferences can also change based on an animal's physiology: newly eclosed female mosquitos feed on nectar, but after mating they prefer blood meals that support the growth of offspring[2,3]. The full extent and nature of metabolic factors that influence food preference in animals are not known[4–6].

Many behaviors of the nematode *Caenorhabditis elegans* are regulated by its bacterial food. In addition to rapid chemotaxis to food-related metabolites and food ingestion through its pharynx[7], it has longer-term foraging behaviors that reflect its evaluation of food quality and quantity over minutes to hours[8]. For example, animals display low-activity dwelling behaviors in high-quality food, and higher-activity roaming behaviors when food is limited[9–12]. At the extreme, animals will abandon a bacterial food source that is hard to eat, toxic, or pathogenic[9,13–18]. In these aversive conditions, *C. elegans* will initially enter the bacterial lawn, but after several hours begins to

exit the lawn and accumulate in food-free regions. The animals also learn to avoid the odor of toxic bacteria in olfactory choice assays[19,20]. These slow acquired aversion behaviors can extend survival by limiting pathogen ingestion and toxin exposure, and may allow animals to select high-quality food sources in their heterogeneous natural environments. They are under extensive neuromodulatory regulation by the biogenic amines serotonin[11,21,22], dopamine[23,24], tyramine[25,26], and octopamine[27], as well as the neuroendocrine cytokine DAF-7[18,28], and the neuropeptides PDF-1 and FLP-1[11,13,16], suggesting that long-term behaviors are guided by slow-acting neuromodulatory systems.

Serotonin links physiology and behavior in many animals. In mammals, serotonin is synthesized by neurons in the brainstem and by enteric neurons and enteroendocrine cells in the digestive system; it regulates locomotion, anxiety, and food responses including appetite, satiety, obesity, gut motility, and nausea[29,30]. Similarly, the serotonergic system in *C. elegans* is implicated in a variety of food-related behaviors. The primary sources of serotonin are three pairs of neurons called NSM, ADF, and HSN. The NSM enteric neurons reside in the pharynx, detect food through proprioceptive endings, and release serotonin to stimulate feeding, locomotor slowing, and dwelling behaviors[31]; NSM also has a key role in acquired aversion to bacteria

[1]Lulu and Anthony Wang Laboratory of Neural Circuits and Behavior, The Rockefeller University, New York, NY, USA. [2]Present address: Department of Molecular Pathobiology, New York University College of Dentistry, New York, NY, USA. [3]Present address: L. E. K. Consulting, New York, NY, USA. ✉e-mail: cori@rockefeller.edu

spiked with mitochondrial toxins[32]. The ADF sensory neurons detect bacterial metabolites and regulate both acute feeding behaviors and learned avoidance of bacterial pathogens[20,33]. The HSN neurons reside in the midbody and regulate egg-laying, locomotion, and dwelling behaviors[34]. Serotonin acts on six receptors with different signaling mechanisms that are collectively expressed on nearly half of all *C. elegans* neurons; several receptors contribute to each serotonin-regulated behavior[11,35]. The general organization of serotonin systems with a few serotonin-expressing neurons, multiple serotonin receptors, and widely distributed serotonin receptor-expressing neurons appears to be shared between mammals and *C. elegans*. As serotonin and other neuromodulators act extrasynaptically, these distributed systems create a challenge for circuit mapping. Understanding information flow requires detailed identification of the functional connections between serotonin- and serotonin-receptor expressing neurons in specific behavioral contexts[35–37].

*Caenorhabditis elegans* feeds on a variety of bacteria in its natural environment[38], but is typically cultivated on *E. coli* in the laboratory[39]. The influences of diet on *C. elegans* growth and physiology have been studied using classical methods[40–44] and extended using the 3983 *E. coli* gene knockout strains in the Keio collection[45], identifying *E. coli* genes required for rapid *C. elegans* development[46], longevity[47], survival under mitochondrial stress[48], and resistance to the toxins 5-Fluoruracil, FUDR, camptothecin, and Aflatoxin B1[49–51]. Here, we use the Keio *E. coli* collection to systematically identify bacterial diets that elicit aversion behavior in *C. elegans*, which we call "mediocre" diets based on this property. Pathway enrichment of these *E. coli* mutants suggests that *C. elegans* monitors dietary quality using a few primary bacterial signals: global bacterial metabolism (*crp*), cysteine biosynthesis, vitamin B6 biosynthesis, ferric iron transport, and bacterial membrane integrity. Using *C. elegans* genetics and circuit mapping, we find that serotonin has a dual role in assessing dietary quality: it promotes aversion from a mediocre diet and suppresses aversion from a high-quality diet through context-dependent roles in serotonin-producing neurons and multiple serotonin receptors. Serotonin acts together with dopamine to modulate octopaminergic and tyraminergic neurons, and serotonin also acts on other neuronal targets to regulate behavior. Our results reveal a distributed neuromodulatory network of biogenic amines, receptors, and target neurons that interpret dietary quality through cooperative and antagonistic effects on behavior.

## Results

### A genome-wide screen for *E. coli* mutants that elicit aversion behavior in *C. elegans*

We evaluated *C. elegans* behavior using a long-term foraging assay that has been widely used to study responses to toxic or pathogenic bacteria[15,16,18,19,52]. 15–20 *C. elegans* L4 larvae are placed on a plate seeded with a dense patch of bacterial food and the location of the animals is tracked by video recording for 20 h (Fig. 1a). Over 80% of animals remain within a patch of the standard *E. coli* bacterial food throughout a 20 h assay, whereas fewer than 10% of animals remain on patches of pathogenic *Pseudomonas aeruginosa* bacteria after 16–20 h (Fig. 1b). An aversion ratio is defined as the steady-state fraction of animals off the food patch 16–20 h after the beginning of the assay (Fig. 1c).

To perform a systematic investigation of bacterial effects on *C. elegans* aversion behavior, we screened the *E. coli* Keio knockout collection[45] of 3983 deletion mutants. Following a prescreen that yielded 262 potentially aversive strains, quantitative video recording identified 22 *E. coli* mutants that elicited significant aversion behavior (Table 1), with an average aversion ratio of 0.4 compared to 0.18 for the parental BW25113 strain (henceforth, wild-type) (Fig. 1c). These 22 genes included multiple hits in five metabolic pathways: regulation of global transcription, biosynthesis of the enterobacterial common

antigen (ECA), biosynthesis of the essential nutrients cysteine and vitamin B6, and ferric iron transport (Fig. 1d, and Supplementary Fig. 1). Aversion ratios for representative genes from each pathway are show in Fig. 1d; the enriched pathways are described briefly below.

In response to reduced availability of a preferred carbon source such as glucose, the *E. coli* adenylate cyclase CyaA catalyzes the production of cyclic AMP (cAMP), which is recognized and bound by CRP, the cAMP receptor protein transcription factor[53–56]. Δ*crp* and Δ*cyaA*, but not other mutants in this pathway, elicited significant aversion in *C. elegans* (Fig. 1d, Supplementary Fig. 1a).

*E. coli* expresses enterobacterial common antigen (ECA) on the outer membrane, where it regulates bacterial cell shape and stress responses. Four ECA biosynthetic mutants, Δ*wzxE*, Δ*rffT*, Δ*rffA* and Δ*rffC*, elicited aversive responses in *C. elegans*. The ECA flippase WzxE flips the resulting ECA unit across the inner membrane prior to ECA maturation[57] (Fig. 1d). No aversion was induced by mutants in earlier or later steps of ECA synthesis, suggesting that accumulation of an inner membrane intermediate, rather than the loss of ECA, drives *C. elegans* aversion behavior (Supplementary Fig. 1b).

In *E. coli*, cysteine is synthesized from serine through the serine acetyltransferase CysE, which converts L-serine to O-acetyl-L-serine (OAS), and the O-acetylserine sulfhydrylase CysK. CysE and CysK form a complex that allosterically activates serine to cysteine conversion[58] (Fig. 1d, and Supplementary Fig. 1c). Δ*cysE* and Δ*cysK* bacteria were moderately aversive to *C. elegans*. Cysteine is not an essential amino acid for *C. elegans* growth[40], but cysteine limitation reduces synthesis of the protective redox metabolite glutathione[59,60].

Four *E. coli* mutants affecting vitamin B6 synthesis elicited *C. elegans* aversion behavior (Fig. 1d, and Supplementary Fig. 1d). Vitamin B6 is an essential nutrient for *C. elegans*[40] and is required for synthesis of serotonin, dopamine, tyramine and octopamine transmitters[61]. In a possible overlap with other aversive bacteria, vitamin B6 is required for bacterial cysteine synthesis *(cysK)* and ECA synthesis *(rffA/wecE)*[62]. *E. coli* vitamin B6 biosynthetic genes are essential for *C. elegans* growth in certain nutrient-limited conditions[63] and can confer *C. elegans* resistance to the toxin 5-fluorouracil[49].

Fifteen enzymes support *E. coli* ferric iron uptake and release through the small-molecule siderophore enterobactin. The four mutants that elicited aversion in *C. elegans*, Δ*fepB*, Δ*fepD*, Δ*fepG*, and Δ*fes* (Fig. 1d, and Supplementary Fig. 1e) accumulate a ferric iron-enterobactin complex in the *E. coli* cytoplasm or periplasm that has been proposed to be toxic to *C. elegans* mitochondria[46,48]. These mutants also elicit production of reactive oxygen species (ROS), but other ROS-producing mutants did not elicit aversion (Supplementary Fig. 1e, f).

Individual mutants that can drive *C. elegans* aversion include Δ*fabH* (fatty acid biosynthesis), Δ*tatB* (twin arginine translocation component, Δ*ynhG* (L,D-transpeptidase), Δ*alaS* (alanine-tRNA ligase), Δ*aceF* (pyruvate dehydrogenase E2 subunit), and Δ*dnaJ* (a chaperone protein) (Supplementary Fig. 1g, and Table 1).

In summary, *C. elegans* shows aversion to several *E. coli* diets with altered metabolic properties. The relatively small number of genes and pathways represented suggests that aversion is an active, relatively specific behavior; in particular, over 200 of the *E. coli* deletion mutants that result in slow *C. elegans* growth did not elicit aversion in this assay[40].

### *E. coli* mutants that elicit *C. elegans* aversion induce stress responses

We selected one *E. coli* mutant from each of the five biological pathways (Δ*crp*, Δ*wzxE*, Δ*pdxJ*, Δ*cysE* and Δ*fepB*) to characterize in more detail. Each selected *E. coli* mutant was tested for effects on *C. elegans* developmental rate, brood size, feeding rate, colonization (a marker of bacterial pathogenesis), and lifespan (Supplementary Fig. 2a). The

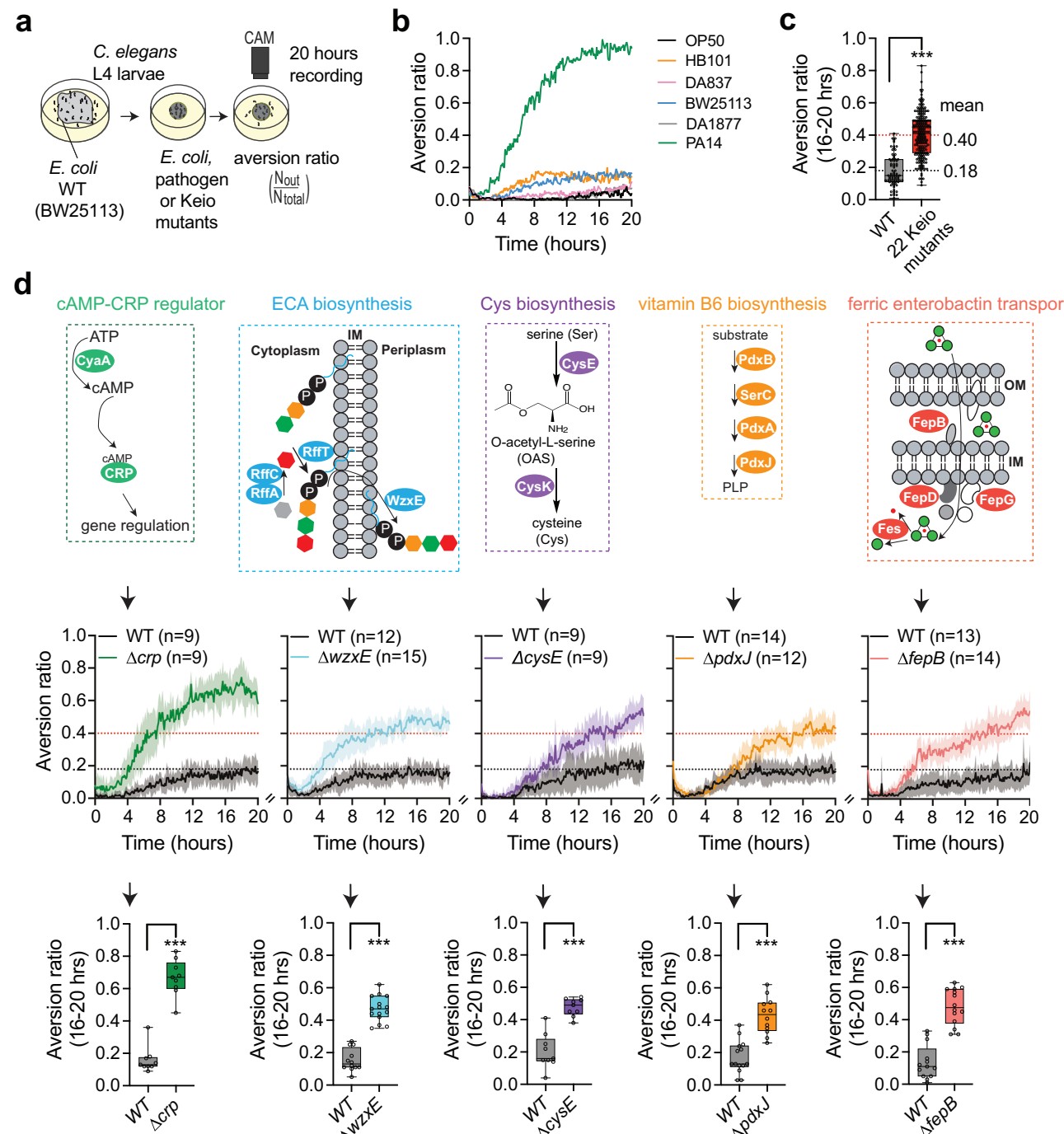

**Fig. 1 | Genome-wide screen for *E. coli* mutants that induce aversion behavior in *C. elegans*.** **a** Schematic depiction of aversion behavior screen. **b** Behavioral responses of wild-type *C. elegans* to different *E. coli* strains and to the pathogenic bacteria *Pseudomonas aeruginosa* PA14. Each trace represents the average of at least 8 individual assays per strain across 20 h. **c** Comparison of pooled behavioral responses of wild-type *C. elegans* on BW25113 (WT) and the 22 Keio mutants identified from the whole-genome screen. The mean values for each group are shown. Black dashed line indicates mean value from WT group (n = 74) and red dashed line indicates mean value from pooled positive hits (n = 229). **d** Simplified depiction of identified metabolic pathways in *E. coli*, full aversion time course for one mutant from each metabolic pathway, and averaged aversion ratios at 16–20 h.

Complete metabolic pathways and additional aversion assays are shown in Supplementary Fig. 1; bacterial growth properties are reported in Supplementary Data 1. Genes identified in this study are highlighted in color. cAMP, cyclic AMP. CRP, cAMP Receptor Protein. ECA, Enterobacterial Common Antigen. PLP, Pyridoxal 5′-phosphate. Ent, Enterobactin. IM, Inner Membrane. OM, Outer Membrane. Each time course trace shows the mean of 9-15 biological replicates; shaded region represents the 95% confidence interval. Each data point in the box plot represents an individual assay (n = 9–15), with median ± quartiles in boxes and Min to Max whiskers. ***P < 0.001 by one-way ANOVA with Dunnett's multiple correction. All sample sizes, statistical tests used, and exact P values are provided in Supplementary Data 4. Source data are provided as a Source Data file.

strongest defects were observed on Δ*fepB*, which caused substantial developmental delay, reduced brood size, and increased colonization of *C. elegans*, in agreement with the toxicity of this strain observed in previous screens[46]. Δ*crp* caused reduced lifespan, reduced feeding,

and a slight developmental delay; Δ*wzxE* also caused reduced feeding and a slight developmental delay; Δ*pdxJ* caused a reduced brood size (Supplementary Fig. 2a). These results indicate that aversive bacterial strains can affect *C. elegans* physiology.

**Table 1 | *E. coli* mutants identified in the *C. elegans* behavioral aversion screen**

| Diet | Gene | Functions |
|---|---|---|
| wild-type | WT (BW25113) | wild-type *E. coli*, K-12 strain |
| mediocre | ***crp***, *cyaA* | cAMP-dependent global carbon response; ***crp*** encodes a DNA-binding transcriptional regulator |
| | ***wzxE***, *rffT*, *rffA*, *rffC* | enterobacterial common antigen (ECA) biosynthesis; ***wzxE*** encodes a lipid IIIECA flippase |
| | ***pdxJ***, *pdxB*, *pdxA*, *serC* | pyridoxine (PN) and pyridoxine 5'-phosphate (PLP) biosynthesis; ***pdxJ*** encodes a PLP synthase |
| | ***cysE***, *cysK* | O-acetyl-L-serine (OAS) and cysteine (Cys) biosynthesis; ***cysE*** encodes a serine acetyltransferase |
| | ***fepB***, *fepD*, *fes*, *fepG* | ferric-enterobactin transport for iron bioavailability; ***fepB*** encodes an ABC transporter periplasmic binding protein |
| | *fabH* | encodes a 3-oxoacyl-[acyl carrier protein] synthase in fatty acid biosynthesis |
| | *tatB* | encodes a twin arginine translocation component inside inner membrane |
| | *ynhG* | encodes a periplasmic L,D-transpeptidase |
| | *alaS* | encodes an alanine-tRNA ligase |
| | *aceF* | encodes a pyruvate dehydrogenase E2 subunit |
| | *dnaJ* | encodes a chaperone protein |

Genes selected for further analysis are highlighted in bold.

To refine this insight, we examined the effects of bacterial mutants on *C. elegans* molecular reporters of physiological stress[14]. Expression of a *daf-7::GFP* reporter in ASJ sensory neurons is induced by bacterial pathogens and by food that is physically difficult for *C. elegans* to ingest[18,28]. This reporter gene was significantly induced by Δ*crp*, Δ*wzxE*, and Δ*fepB* bacteria, but not by Δ*cysE* or Δ*pdxJ* bacteria (Fig. 2a). An *hsp-6::GFP* reporter for mitochondrial stress was also significantly induced by Δ*crp* and Δ*fepB* bacteria[46] (Fig. 2b). A *gst-4::GFP* reporter for oxidative stress was induced by Δ*cysE* and Δ*wzxE* bacteria (Fig. 2c), but not by Δ*crp*, Δ*fepB*, or Δ*pdxJ* bacteria. These results suggest that aversive diets can selectively activate *C. elegans* stress responses. Several other *C. elegans* stress reporters were also induced by one or more of the aversive diets (Supplementary Fig. 2b-f).

The metabolic effects of the bacterial mutants were confirmed by chemical supplementation either prior to bacterial growth (pre-adding) or immediately before the aversion assay (post-adding) (Fig. 2d). Addition of glucose to the assay plates during bacterial growth, which should bypass the metabolic function of CRP, fully suppressed *C. elegans* aversion to the Δ*crp* diet (Fig. 2d). Addition of cysteine or its precursor *O*-acetyl-serine either before or after bacterial growth suppressed *C. elegans* aversion to the Δ*cysE* diet, consistent with a cysteine-related dietary deficiency (Fig. 2d). Addition of vitamin B6 to Δ*pdxJ* and related bacterial mutants suppressed *C. elegans* aversion if the vitamin was provided before bacterial growth, but did not suppress aversion if only provided during the assay (Supplementary Fig. 2g). This pattern suggests that aversion results in part from altered bacterial metabolism[49].

Foraging in *C. elegans* is sensitive to bacterial density[64,65]. To control for effects of differential bacterial growth, we measured and varied the density of wild-type, Δ*crp*, and Δ*cysE* bacterial strains and assessed effects on *C. elegans* behavioral responses (Fig. 2e and Supplementary Fig. 3). Attraction to wild-type bacteria was stable across a 10-fold range of cell densities. Aversion to Δ*crp* was observed across cell densities, consistent with a stress response. Aversion to Δ*cysE* was rescued at high bacterial densities, and was also rescued by acutely supplementing low Δ*cysE* bacterial densities with cysteine (Fig. 2e and Supplementary Fig. 3). These results are consistent with a cysteine-related metabolic deficit.

The results of these experiments suggest that at least two different bacterial qualities can drive *C. elegans* aversion: a metabolic change that induces mitochondrial stress (Δ*crp*, Δ*fepB*) and a metabolic change that induces oxidative stress (Δ*cysE*); both processes may contribute to aversion from Δ*wzxE*. Δ*crp* and Δ*cysE* were used as representative examples of aversive bacteria of two different classes. We refer to these as "mediocre" diets based on their aversive qualities and their activation of *C. elegans* stress reporters.

## *C. elegans* evaluates opposing food qualities with distinct classes of serotonergic neurons

To ask how *C. elegans* distinguishes between bacteria in the aversion assay, we began with serotonin, a neuromodulator that regulates multiple food-related behaviors as well as avoidance of pathogenic bacteria[11,14,20,32,66]. Animals mutant for the tryptophan hydroxylase gene *tph-1* lack all neuronally-synthesized serotonin[67,68]. We found that two independent *tph-1* mutants had reduced aversion to both Δ*crp* and Δ*cysE* mediocre diets, while retaining apparently normal responses to wild-type bacteria (Fig. 3a and Supplementary Fig. 4a).

Three classes of *tph-1*-expressing neurons synthesize serotonin in *C. elegans* hermaphrodites. To ask which neurons regulate aversion, we eliminated *tph-1* in individual neuron classes using Cre/lox recombination[25], and tested the resulting animals on wild-type, Δ*crp* and Δ*cysE* diets. Remarkably, each of the three serotonin-producing neurons affected aversion behaviors (Fig. 3a). Depletion of serotonin synthesis in ADF neurons resulted in a small but significant increase in aversion from wild-type bacteria; depletion of serotonin synthesis in NSM neurons decreased aversion from both Δ*crp* and Δ*cysE* mediocre diets; and depletion of serotonin synthesis in HSN neurons decreased aversion from the Δ*crp* diet. These results suggest that serotonin can either increase or decrease aversion behavior, depending on the neuronal serotonin source and the dietary context.

To validate these results with an orthogonal approach, we manipulated individual serotonergic neurons and tested the resulting strains on the three bacterial diets. Acute chemogenetic silencing of ADF neurons using a histamine-gated chloride channel (HisCl1) transgene[69] and exogenous histamine significantly increased aversion from wild-type bacteria (Fig. 3b), resembling the *tph-1* knockout in ADF. Acute chemogenetic silencing of NSM neurons suppressed aversion to the mediocre Δ*crp* and Δ*cysE* diets, recapitulating the effects of the *tph-1* knockout in NSM (Fig. 3c). Finally, a strain in which the HSN neurons were genetically ablated (*egl-1(gf)*) had diminished aversion to the mediocre Δ*crp* diet, resembling animals in which HSN did not synthesize serotonin (Fig. 3d). These results confirm that serotonergic ADF neurons prevent aversion from wild-type diets, whereas serotonergic NSM and to a lesser extent HSN neurons promote aversion from mediocre Δ*crp* and Δ*cysE* diets (Fig. 3e).

## The serotonin receptor SER-5 suppresses aversion from a wild-type diet together with the biogenic amine octopamine

*C. elegans* has six known serotonin receptors: two serotonin-gated ion channels (MOD-1, a chloride channel, and LGC-50, a cation channel), and four G-protein coupled receptors (SER-5 (Gs-coupled), SER-7 (Gs-coupled), SER-1 (Gq-coupled) and SER-4 (Gi/o-coupled))

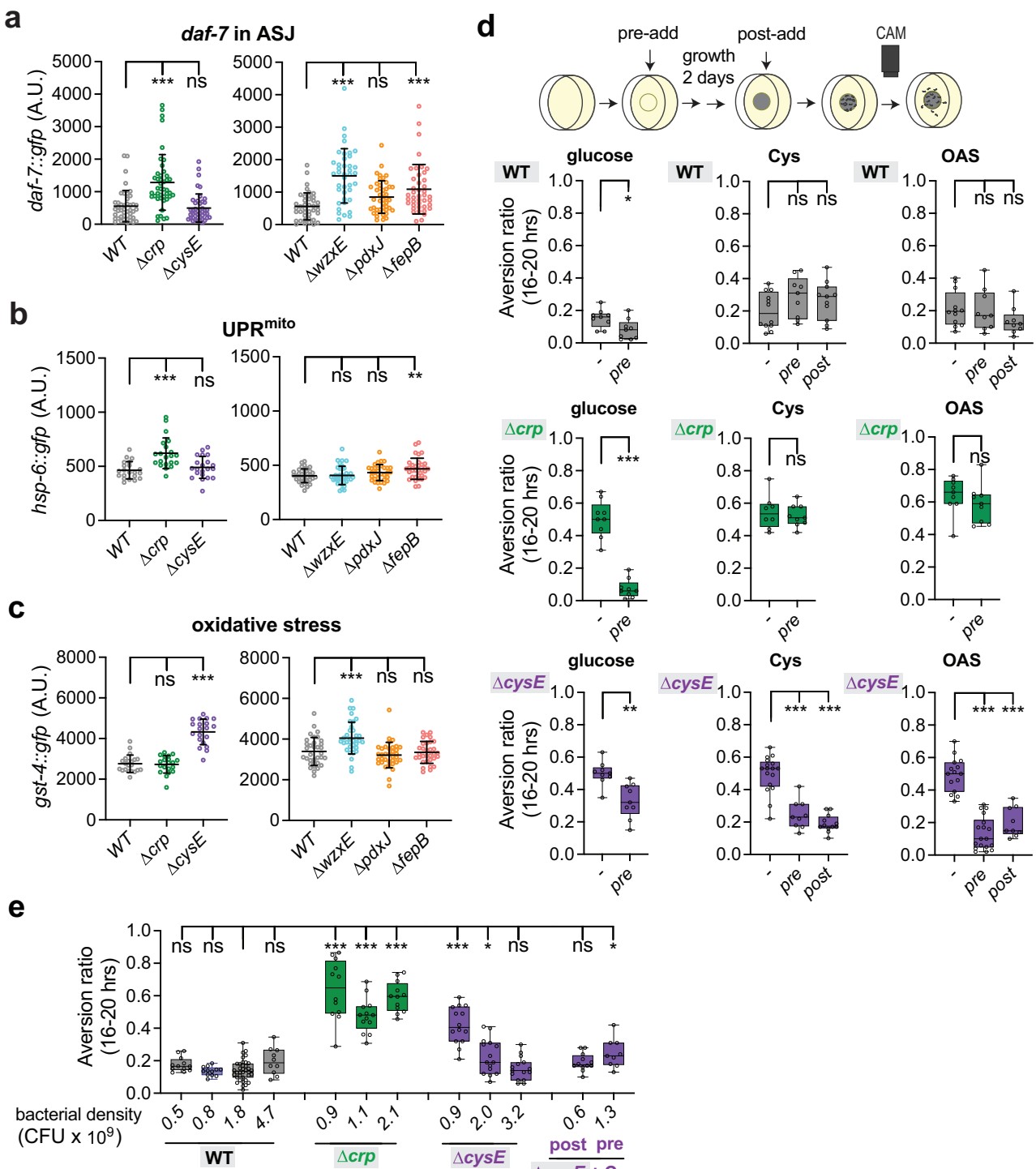

**Fig. 2 | Diets that induce aversion elicit stress responses in *C. elegans*. a–c** Effects of mediocre bacterial diets on *C. elegans* stress reporters. **a** Quantification of *daf-7::gfp* marker in ASJ sensory neurons. **b** Quantification of the mitochondrial UPR marker *hsp-6::gfp*. **c** Quantification of the oxidative stress marker *gst-4::gfp*. GFP fluorescence intensities quantified as Arbitrary Units (A.U.). Each data point indicates fluorescence from an individual animal (n = 20-45). Error bars indicate mean ± SD. ns, not significant, **P < 0.01, ***P < 0.001 by one-way ANOVA corrected by Dunnett's multiple comparisons. **d** Effects of chemical supplementation on aversion behavior to wild-type, Δ*crp*, or Δ*cysE* bacteria. Top, schematic diagram of chemical supplementation. Pre-adding indicates chemicals provided during bacteria growth; post-adding indicates chemicals supplied one hour before the behavioral assay. Final concentrations in assay plates were 0.4% glucose, 200 μM cysteine (pre-adding) or 50 μM cysteine (post-adding), and 200 μM OAS. Each data point indicates individual

assay (n = 8–18 assays). Results are shown with median ± quartiles in boxes and Min to Max whiskers. ns, not significant, *P < 0.05, **P < 0.01, ***P < 0.001 by two-tailed, unpaired t test or one-way ANOVA corrected by Dunnett's multiple comparisons. **e** Aversion behaviors at different bacterial densities. Wild-type, Δ*crp*, or Δ*cysE* were seeded at different densities and CFU from control plates were counted at the beginning of the behavioral assay for most assays; for pre- and post- added cysteine, CFU from control plates were counted at the end of the assay. Each data point indicates individual assay (n = 10-36). Results are shown with median ± quartiles in boxes and Min to Max whiskers. ns, not significant, *P < 0.05, ***P < 0.001 by one-way ANOVA corrected by Dunnett's multiple comparisons. Additional stress reporters are shown in Supplementary Fig. 2, and additional assays and bacterial counts are shown in Supplementary Fig. 3. All sample sizes, statistical tests used, and exact P values are provided in Supplementary Data 4. Source data are provided as a Source Data file.

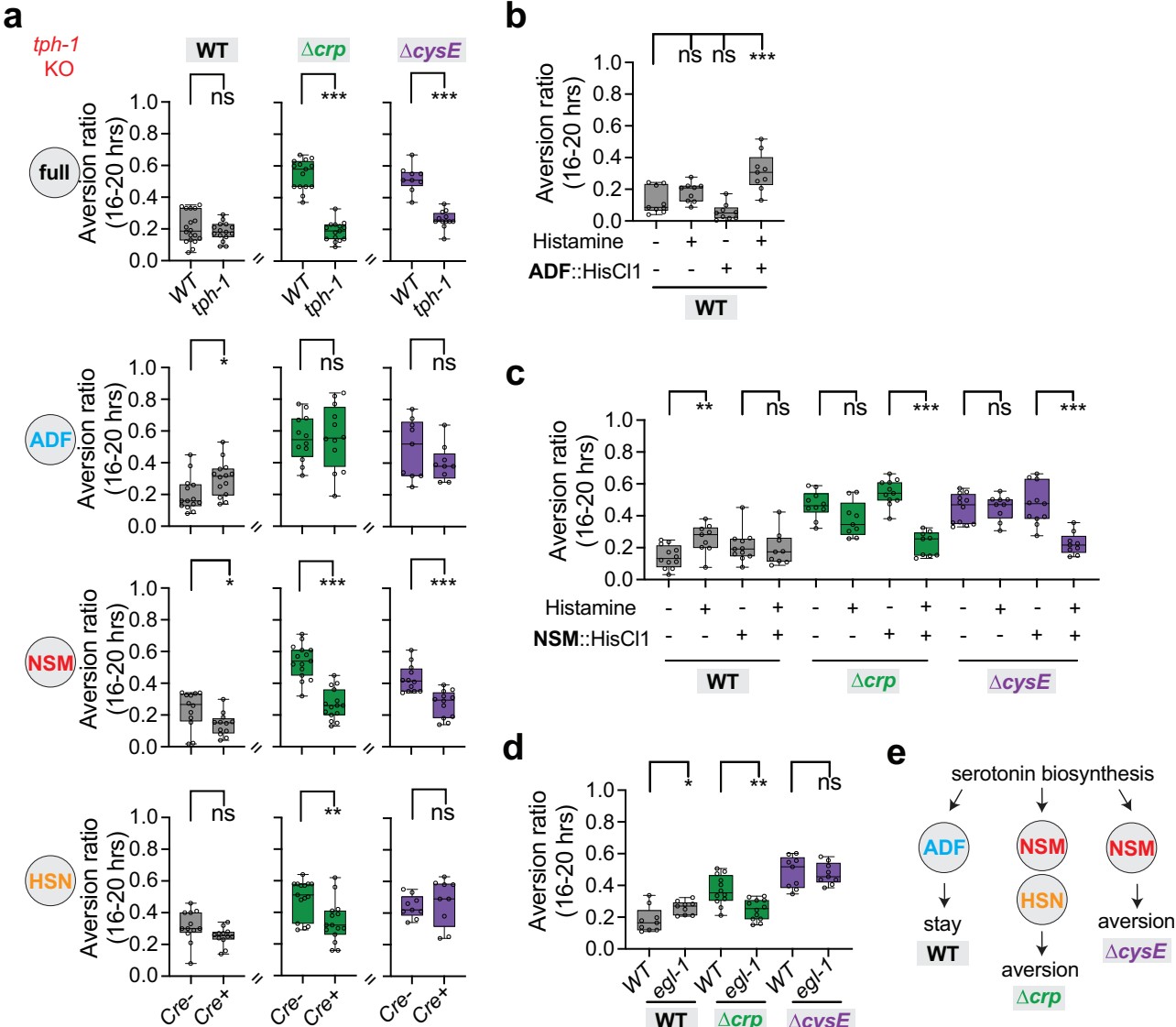

**Fig. 3 | *C. elegans* evaluates opposing food qualities through different serotonergic neurons. a** Aversion behaviors of *tph-1* null mutants and neuron-specific knockout mutants on three diets. *tph-1* is expressed in three major serotonergic neuron classes, ADF, NSM and HSN, and encodes a tryptophan hydroxylase that catalyzes the first step in serotonin biosynthesis. Cell-specific knockout of a single-copy genomic *tph-1* insertion was achieved by Cre-lox recombination[11] (n = 9–15 assays). All figures show the *tph-1(mg280)* null allele, other than Supplementary Fig. 4, which shows similar aversion behaviors in the *tph-1(n4622)* null allele. **b** ADF silencing with a HisCl transgene and histamine induces aversion to wild-type *E. coli* BW25113 (n = 9 assays). **c** NSM silencing with a HisCl transgene and histamine

suppresses aversion to Δ*crp* and Δ*cysE* diets (n = 9–11 assays). **d** HSN-deficient *egl-1*(gain-of-function) mutants show decreased aversion to the Δ*crp* diet (n = 9–12 assays). **e** Schematic model of serotonergic neuron functions. For all panels, each data point indicates individual assay. Results are shown with median ± quartiles in boxes and Min to Max whiskers. ns, not significant, *P < 0.05, **P < 0.01, ***P < 0.001 by two-tailed, unpaired t test (panels **a**, **c**, **d**) or one-way ANOVA corrected by Dunnett's multiple comparisons (**b**). Full genotypes of all *C. elegans* strains are provided in Supplementary Data 3. All sample sizes, statistical tests used, and exact P values are provided in Supplementary Data 4. Source data are provided as a Source Data file.

(Fig. 4a)[67]. Collectively, these receptors are expressed in almost half of all neuronal classes, and typically act cooperatively in foraging and feeding behaviors[35]. We tested mutants for each serotonin receptor for aversion on the wild-type diet as well as mediocre Δ*crp* and Δ*cysE* diets; for clarity, the results are discussed separately on each diet below.

Two of the six serotonin receptor mutants, *ser-4* and *ser-5*, showed strikingly enhanced aversion to the wild-type diet, while the mutants in the other four receptor genes were normal (Fig. 4b). Previous studies suggested that serotonin release from ADF neurons activates the excitatory SER-5 receptor in a variety of neurons to regulate sensory responses, food intake, and longevity[70–72]. *ser-5* is expressed in RIC neurons[35,73], and serotonin released from ADF can activate RIC[74,75]. We

found that a *ser-5* cDNA rescued behavior on the wild-type diet when expressed from its endogenous promoter or from promoters for *tdc-1* (expressed in RIM and RIC neurons) or *tbh-1* (expressed in RIC neurons) (Fig. 4c). Expression in *ser-5*-expressing enteric neurons did not rescue behavior on the wild-type diet (Fig. 4c).

The RIM and RIC neurons express tyrosine decarboxylase (*tdc-1*) to produce the neurotransmitter tyramine, and RIC also expresses tyramine beta-hydroxylase (*tbh-1*) that converts tyramine into octopamine. *tdc-1* mutants lack both tyramine and octopamine, whereas *tbh-1* mutants lack octopamine and accumulate excess tyramine[33]. We examined the behaviors of *tdc-1* and *tbh-1* mutant animals on wild-type diets. Like *ser-5* mutant animals, both *tdc-1* and *tbh-1* mutants had substantial aversion from

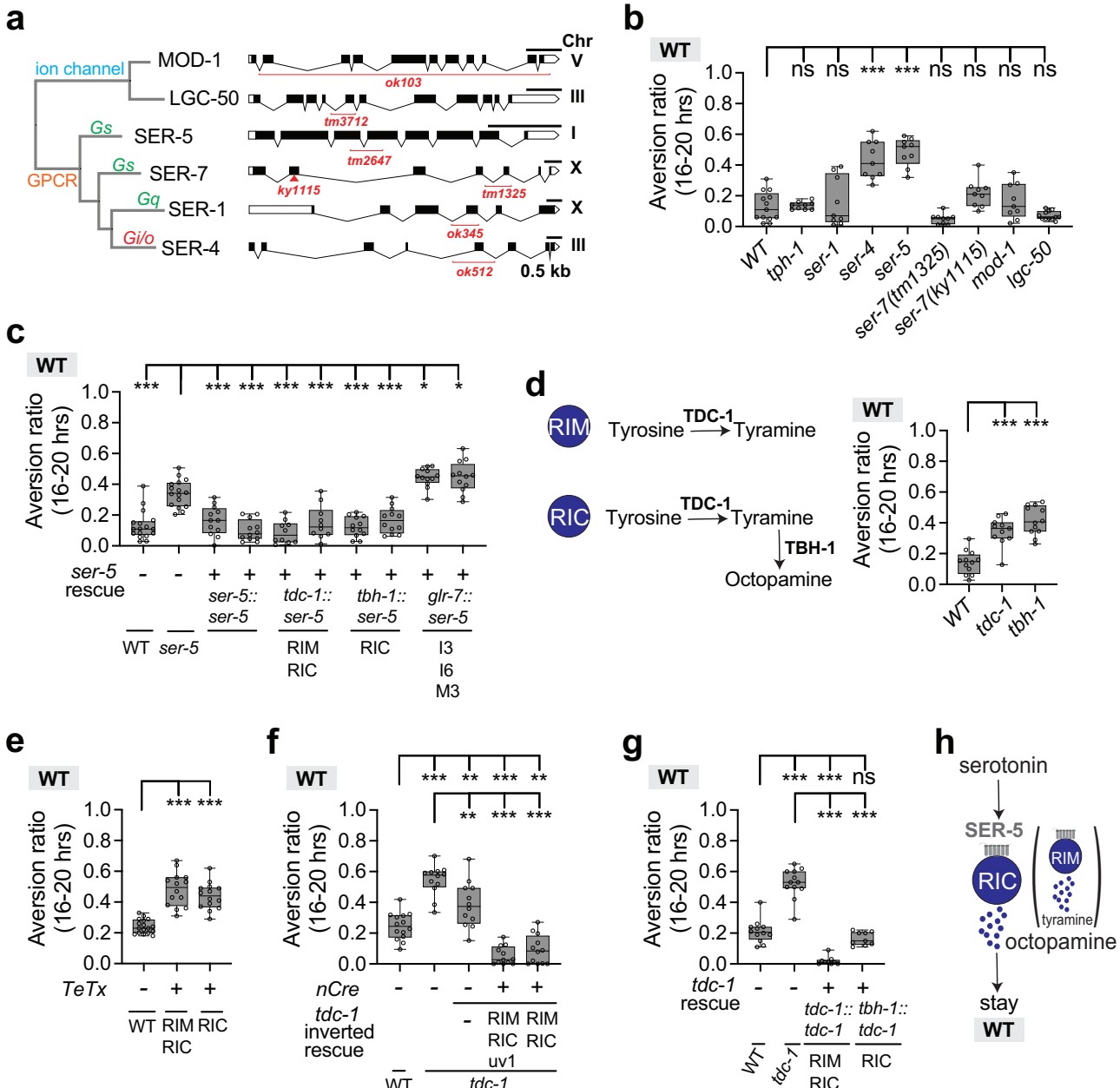

**Fig. 4 | The serotonin receptor SER-5 suppresses aversion from a wild-type diet.**
**a** Schematic diagram of six serotonin receptors in *C. elegans*, with relevant
G-protein for GPCRs indicated. Serotonin receptor mutations are indicated for each
gene. **b** Behavioral responses of wild-type, *tph-1*, and serotonin receptor mutant
animals on the wild-type BW25113 diet. *ser-4* and *ser-5* mutants showed significant
aversion (n = 9–12 assays). **c** Aversion behavior of *ser-5* mutants expressing *ser-5*
cDNA from its endogenous promoter (*ser-5*) or promoters for RIM and RIC (*tdc-1*),
RIC (*tbh-1*), or enteric neurons of the pharynx (*glr-7*). Two transgenic lines were
tested for each rescue plasmid (n = 10–17 assays). **d** Aversion behavior of *tdc-1* and
*tbh-1* null mutants on BW25113 (n = 11–12 assays). TDC-1 tyrosine decarboxylase is
expressed in RIM and RIC neurons and uv1 neuroendocrine cells, and converts
tyrosine to tyramine (TA). TBH-1 tyramine b-hydroxylase converts tyramine to
octopamine (OA) in RIC neurons and gonadal sheath cells. **e** Expression of the

tetanus toxin light chain in RIM and/or RIC neurons to block synaptic release
induced aversion to BW25113 (n = 14–17 assays). **f** Rescue of *tdc-1* using an inverted
Cre-lox strategy. *Cre* expression was driven by the *rimb-1* promoter (intersection in
RIM, RIC, and uv1 cells), or *dmsr-7* promoter (intersection in RIM and RIC neurons)
(n = 12–14 assays). **g** Rescue of *tdc-1* mutants by expression of *tdc-1* cDNA from the
*tdc-1* promoter (RIM, RIC and uv1 cells) or the *tbh-1* promoter (RIC and gonadal
sheath cells) (n = 9–12 assays). **h** Schematic model of SER-5 function. For **b**–**g** each
data point indicates individual assay. Results are shown with median ± quartiles in
boxes and Min to Max whiskers. ns, not significant, *P < 0.05, **P < 0.01, ***P < 0.001
by one-way ANOVA corrected by Dunnett's (**b**–**e**) or Tukey's (**f**, **g**) multiple com-
parisons. Full genotypes of all *C. elegans* strains are provided in Supplementary
Data 3. All sample sizes, statistical tests used, and exact P values are provided in
Supplementary Data 4. Source data are provided as a Source Data file.

the wild-type diet (Figs. 4d, 7g). Expressing the tetanus toxin light
chain to block synaptic release from RIC neurons or from RIM and
RIC neurons also induced aversion (Fig. 4e). Normal behavior was
restored to *tdc-1* mutants by restoring expression in RIC neurons
or in RIC and RIM neurons (Fig. 4f, g).

Together, these results suggest that serotonin from ADF acts on
SER-5 to enhance the activity of tyramine and octopamine-producing
cells that suppress aversion from a high-quality diet (Fig. 4h). The
combination of strong *ser-5* rescue in RIC alone (Fig. 4c) and equivalent
levels of aversion induced by *tbh-1* and *tdc-1* mutations (Fig. 4d) suggest

**a**

Δ*crp*

Aversion ratio (16-20 hrs)

*** ns ns ** *** *** * ***

WT *tph-1* *ser-1* *ser-4* *ser-5* *ser-7(tm1325)* *ser-7(ky1115)* *mod-1* *lgc-50*

**b**

Δ*crp*

Aversion ratio (16-20 hrs)

*** ns *** *** *** ***

*ser-7* rescue: − − − + + + +

WT *tph-1* *ser-7*    *ser-7:: ser-7*    *flp-21:: ser-7*

RMG PQR M2 PVW    RMH I1 NSM

**c**

Δ*crp*

Aversion ratio (16-20 hrs)

*** *** * *** ***

*** ** *** *** *** * ns ns ns ns ns ns ns ns ns ns ns *** ns

*ser-7* rescue: − − + + + + + + + + + + + + + + + + +

WT *ser-7*    *ceh-45:: ser-7*  *lgc-8:: ser-7*    *del-7:: ser-7*  *ceh-28:: ser-7*  *ceh-19:: ser-7*      *ceh-53:: ser-7*  *flp-15:: ser-7*

MI I1 I3    I1    NSM    M4    MC    MC M4    NSM M4    M4 I5    I2

I1 neuron related    pharyngeal pumping related or control

**d**

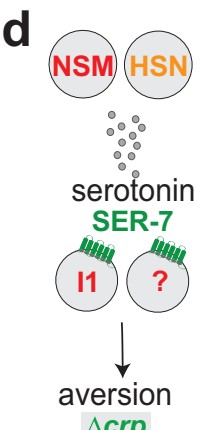

NSM  HSN

serotonin

SER-7

I1  ?

aversion

Δ*crp*

that octopamine from RIC neurons is the primary signal that suppresses aversion from wild-type bacteria (Fig. 4h) but tyramine may also have a role (see below).

**The serotonin receptor SER-7 and dopamine promote aversion from a mediocre Δ*crp* diet**

The serotonin receptor mutants *ser-7, mod-1*, and *lgc-50* had reduced aversion from the mediocre Δ*crp* diet (Fig. 5a). Previous studies

showed that the excitatory SER-7 receptor acts in enteric neurons of the pharynx to regulate feeding[10]. We found that a *ser-7* cDNA rescued aversion on the mediocre Δ*crp* diet when expressed from its endogenous promoter or a *flp-21* promoter that is expressed in overlapping enteric neurons (Fig. 5b)[10]. Among these neurons, *ser-7* expression in the I1 enteric neuron partly rescued aversion from the mediocre Δ*crp* diet, but expression in several other feeding-related neurons did not rescue (Fig. 5c). These results suggest that the I1 neuron, likely

**Fig. 5 | The serotonin receptor SER-7 drives aversion from the mediocre Δcrp diet. a** Behavioral responses of wild-type, *tph-1*, and serotonin receptor mutant animals on the mediocre Δcrp diet. Aversion is diminished in two *ser-7* null mutants and in *mod-1* and *lgc-50* mutants (n = 8–24 assays). The *ser-7(tm1325)* allele was used in all subsequent experiments. **b** Aversion behavior of *ser-7* mutants expressing *ser-7* cDNA from its endogenous promoter (*ser-7*), or a promoter for overlapping neurons (*flp-21*). Two transgenic lines were tested for each plasmid (n = 9–27 assays). **c** Aversion behavior of *ser-7* mutants with *ser-7* rescue in the I1 neuron and other sets of enteric neurons. Two transgenic lines were tested for each rescue plasmid (n = 9–73 assays). **d** Schematic model of SER-7 serotonin signaling in aversion from the mediocre Δcrp diet. For (**a**–**c**), each data point indicates individual assay. Results are shown with median ± quartiles in boxes and Min to Max whiskers. ns, not significant, *P < 0.05, **P < 0.01, ***P < 0.001 by one-way ANOVA corrected by Dunnett's (**a**, **b**) or Tukey's (**c**) multiple comparisons. Full genotypes of all *C. elegans* strains are provided in Supplementary Data 3. All sample sizes, statistical tests used, and exact P values are provided in Supplementary Data 4. Source data are provided as a Source Data file.

responding to serotonin released from the NSM neurons, is part of an enteric circuit that promotes aversion from the mediocre Δcrp diet (Fig. 5d). *ser-7* acts in the MC and M4 enteric neurons to promote feeding or recognition of familiar food[10,21], but expression in these neurons did not rescue aversion to the mediocre Δcrp diet, nor did expression in I1 neuron rescue *ser-7* feeding, indicating that aversion and feeding are separable functions of *ser-7* (Fig. 5c, Supplementary Fig. 4b). Additional *ser-7*, *mod-1*, and *lgc-50* -expressing neurons that affect aversion from Δcrp remain to be identified.

Further insight into aversion from the Δcrp diet came from examination of the neurotransmitter dopamine. Animals mutant for the dopamine biosynthetic enzyme *cat-2* (tyrosine-3-monooxygenase) had diminished aversion to the Δcrp mediocre diet (Fig. 6a). *cat-2* is expressed in three classes of ciliated sensory neurons, CEP, ADE, and PDE. Expression of a *cat-2* cDNA from its own promoter or a promoter that was selectively expressed in CEP sensory neurons rescued aversion on the Δcrp diet (Fig. 6a).

Among dopamine receptors, animals mutant for the inhibitory (Gi/o-coupled) dopamine receptor DOP-6, but not other dopamine receptors, had reduced aversion to the Δcrp mediocre diet, like *cat-2* mutants (Fig. 6b). *dop-6* is expressed in RIC neurons[35,73], and previous studies showed that *dop-6* can inhibit RIC activity[72]. We found that expression of a *dop-6* cDNA in the RIC neuron, or in RIM and RIC neurons, rescued aversion from the mediocre Δcrp diet (Fig. 6c).

To further explore the dopamine circuit, we used double mutants to ask how dopamine interacts with RIM and RIC neurotransmitters. Animals mutant for the RIM/RIC neurotransmitter enzymes *tdc-1* or *tbh-1* avoided the mediocre Δcrp diet (Figs. 6d, 7h), as did *cat-2 tdc-1* and *cat-2 tbh-1* double mutants, unlike *cat-2* single mutants (Fig. 6d). *tdc-1 dop-6* or *tbh-1 dop-6* double mutants also avoided the mediocre Δcrp diet, as expected if DOP-6 inhibits tyramine and octopamine function from RIM and RIC (Fig. 6e). As RIC expression was sufficient to rescue *dop-6*, and *cat-2* aversion defects were fully suppressed by *tbh-1*, it is likely that octopamine from RIC neurons can suppress aversion from the Δcrp diet but is inhibited by dopamine.

The CEP neurons that produce dopamine are implicated in mechanosensory detection of bacteria[76]. To ask whether Δcrp bacteria might have unusual mechanosensory features, we examined these bacteria by scanning electron microscopy. Under our growth conditions, Δcrp cells were significantly larger than wild-type *E. coli* (Fig. 6f). A second bacterial strain with morphological alterations was ΔwzxE, the ECA biosynthesis mutant (Fig. 6f). These defects may be related to the defective outer membrane structures of ΔwzxE mutants. Like aversion to the Δcrp diet, aversion from the ΔwzxE diet required *tph-1*, *cat-2*, and *ser-7* genes (Supplementary Fig. 4c).

In summary, aversion from the mediocre Δcrp diet involves antagonistic interactions between serotonin and dopamine, which promote aversion, and octopamine, which suppresses aversion. Cell-specific rescue experiments identify NSM, HSN, CEP, I1, and RIC neurons as regulators of Δcrp aversion.

### Two serotonin receptors, SER-1 and MOD-1, promote aversion from a mediocre ΔcysE diet

The serotonin receptor mutants *ser-1* and *mod-1* had reduced aversion from the mediocre ΔcysE diet, like *tph-1* mutants (Fig. 7a, b). The excitatory receptor *ser-1* is expressed in RIA and RIC neurons, among other locations[35,73,74,77], and regulates nociception by acting in RIC neurons[74]. We found that a *ser-1* cDNA rescued aversion from the ΔcysE diet when expressed from its endogenous promoter or when expressed specifically in RIA neurons, but did not rescue when expressed in RIC neurons (Fig. 7c).

Serotonin released from NSM can inhibit RIM and RIC neurons through the MOD-1 receptor to modulate locomotion or feeding[78–80]. Expressing *mod-1* in RIC neurons or in RIM and RIC neurons rescued aversion of *mod-1* mutants on the mediocre ΔcysE diet (Fig. 7d), Double mutants between *mod-1* and the RIM/RIC neurotransmitter enzymes *tdc-1* or *tbh-1* restored aversion on the mediocre ΔcysE diet (Fig. 7e). The combination of *mod-1* rescue in RIC neurons and aversion in *mod-1 tbh-1* double mutants indicates that octopamine from RIC can suppress aversion from the ΔcysE diet, but is inhibited by *mod-1*.

Notably, different combinations of genes, neurons, and bacterial diets did not always yield simple interpretations. For example, *tdc-1 ser-1* double mutants had strong aversion to the ΔcysE diet, indicating that *tdc-1* can suppress *ser-1*, even though *ser-1* aversion was not rescued in the RIC neurons (Fig. 7f). The dopamine-deficient *cat-2* mutant had reduced aversion to the mediocre ΔcysE diet, but CEP neurons and the dopamine receptor *dop-6* were not specifically required for this behavior (Supplementary Fig. 5a, b), further distinguishing the requirements for aversion from Δcrp and ΔcysE mediocre diets.

Finally, genetic interactions between *tph-1*, *tdc-1*, and *tbh-1* highlighted the context-dependent contributions of serotonin, tyramine, and octopamine on different bacterial diets (Fig. 7g-i). Double mutants between *tph-1* and *tbh-1* resembled *tph-1* mutants on all diets, pointing to a central role of serotonin across conditions, and a secondary role for octopamine. However, on a wild-type diet tyramine may have at least some serotonin-independent functions, as *tdc-1 tph-1* double mutants had intermediate phenotypes compared to either single mutant (Fig. 7g).

## Discussion

Ingestion of toxic foods that induce sickness, nausea, or gastrointestinal dysfunction is followed by aversion behaviors that prolongs an animal's survival[1,81]. *C. elegans* initially enters a bacterial lawn using a simple rule-of-thumb based on bacterial cell density[64] but after several hours of exposure animals avoid lethal bacterial foods such as certain RNAi-expressing bacteria[14], food spiked with exogenous toxins[14,32], or pathogens[15,16,18,82]. Here we show that a less extreme, but related aversion behavior is induced by a small number of mutant *E. coli* diets that induce mitochondrial or redox stress markers. We find that aversion is regulated by interacting serotonin, dopamine, and octopamine neurotransmitters and receptors.

Our genome-wide screen identified 22 *E. coli* mutants that induced aversive responses, a small fraction of the 244 *E. coli* mutants that result in *C. elegans* developmental delay[46]. Aversion is a stringent assay that requires animals to spend a substantial fraction of their time away from the only available food, and thus it may not be induced by simple nutrient limitation. In addition, the aversion assay begins at the L4 larval stage, when animals may have sufficient reserves of many nutrients. In this setting, we suggest that aversion represents a behavioral response to diets that induce metabolic stress. In agreement

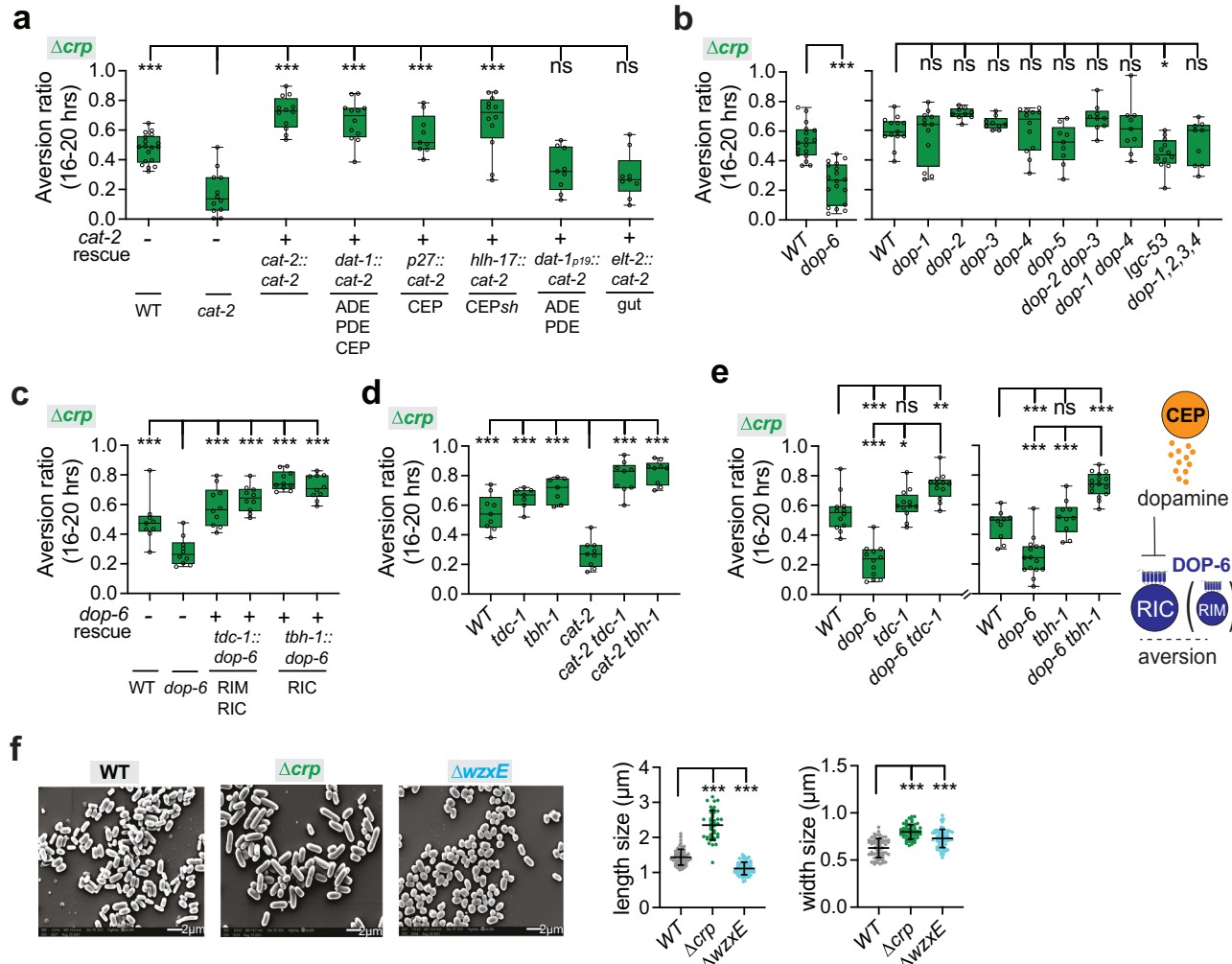

**Fig. 6 | Aversion from the mediocre Δ*crp* diet requires dopamine. a** Aversion behavior of *cat-2* mutants expressing *cat-2* genomic DNA from its the endogenous promoter (*cat-2*), or promoters for all dopaminergic neurons (*dat-1*), CEP neurons (*p27*), CEP sheath glial cells (*hlh-17*), ADE and PDE neurons (*dat-1p19*), or intestine (*elt-2*) on the mediocre Δ*crp* diet (n = 9–18 assays). **b** Aversion behavior of WT and dopamine receptor mutants on the mediocre Δ*crp* diet (n = 9–18 assays). **c** Aversion behavior of *dop-6* mutants with *dop-6* rescue in *tdc-1*- or *tbh-1*- expressing cells on the mediocre Δ*crp* diet (n = 9-10 assays). **d** Genetic interaction of *cat-2* with *tdc-1* or *tbh-1* (tyramine and/or octopamine biosynthesis mutants) on the mediocre Δ*crp* diet (n = 7-9 assays). **e** Genetic interaction of *dop-6* with *tdc-1* or *tbh-1* on the mediocre Δ*crp* diet, and schematic model of dopamine signaling. Stopped and dotted lines represent inhibition of RIC and octopamine (n = 10–15 assays).

**f** Scanning electron micrographs of wild-type BW25113 bacteria, Δ*crp* bacteria, and Δ*wzxE* bacteria (left) and quantification of bacterial length and width (right, with SEM). For length, WT, n = 58; Δ*crp*, n = 41; Δ*wzxE*, n = 72. For width, WT, n = 59; Δ*crp*, n = 54; Δ*wzxE*, n = 60. Scale bar, 2 μm. For (**a**–**e**), each data point indicates individual assay. Results are shown with median ± quartiles in boxes and Min to Max whiskers. ns, not significant, *P < 0.05, **P < 0.01, ***P < 0.001 by one-way ANOVA corrected by Dunnett's multiple comparisons (**a**–**c**) or Tukey's multiple comparisons (**d**, **e**), or by two-tailed, unpaired t test (**b**, *dop-6*). For (**f**), each data point indicates individual bacterial cell. ***P < 0.001 by two-tailed Mann-Whitney test. Full genotypes of all *C. elegans* strains are provided in Supplementary Data 3. All sample sizes, statistical tests used, and exact P values are provided in Supplementary Data 4. Source data are provided as a Source Data file.

with this interpretation, the genes identified here overlap with those that elicit avoidance of mitochondrial toxins or bacterial pathogens[14,16,32,83].

We focused our studies on two bacterial genes whose effects on *C. elegans* had not been examined previously, the catabolite regulator CRP and the cysteine biosynthesis gene CysE. CRP is a global regulator of bacterial metabolism on low-carbohydrate media. On nematode growth medium, Δ*crp* bacteria were abnormally large, and induced expression of *C. elegans* stress reporters associated with toxic or inedible bacteria (*daf-7*) and the mitochondrial unfolded protein response *(hsp-6)*[84]. These two reporters were also induced by the Δ*fepB* ferric enterobactin transport mutant that elicited aversion. Interestingly, our screen yielded the same four *fes/fep* genes as screens for *C. elegans* developmental delay[46], poor development on a nutrient-limited food source[63], and synthetic lethality with the *C. elegans* mitochondrial mutant *spg-7*[48]. The ferric-enterobactin complex that

accumulates in these strains appears to cause mitochondrial toxicity[48]. As mitochondrial toxins can drive food aversion in *C. elegans*[14], we suggest that mitochondrial dysfunction contributes to aversion to Δ*fes/fep* and Δ*crp* diets. Aversion to the mediocre Δ*crp* diet may also reflect an abnormal size or shape that makes the bacteria harder to ingest; hard-to-ingest food can also drive behavioral aversion and *daf-7* stress reporters[28]. The identification of *cysE* as a mediocre food that elicits the oxidative stress reporter *gst-4* (i.e., glutathione S-transferase) is consistent with the role of cysteine in synthesis of the antioxidant glutathione. *C. elegans* synthesizes glutathione from cysteine and other precursors, and this pathway requires more cysteine than protein synthesis. Either an excess or a deficiency of dietary thiols has effects on redox states that are disadvantageous for *C. elegans*[85], explaining why cysteine, an amino acid that *C. elegans* can synthesize on its own, nonetheless can be limited by bacterial production and disruption of glutathione homeostasis.

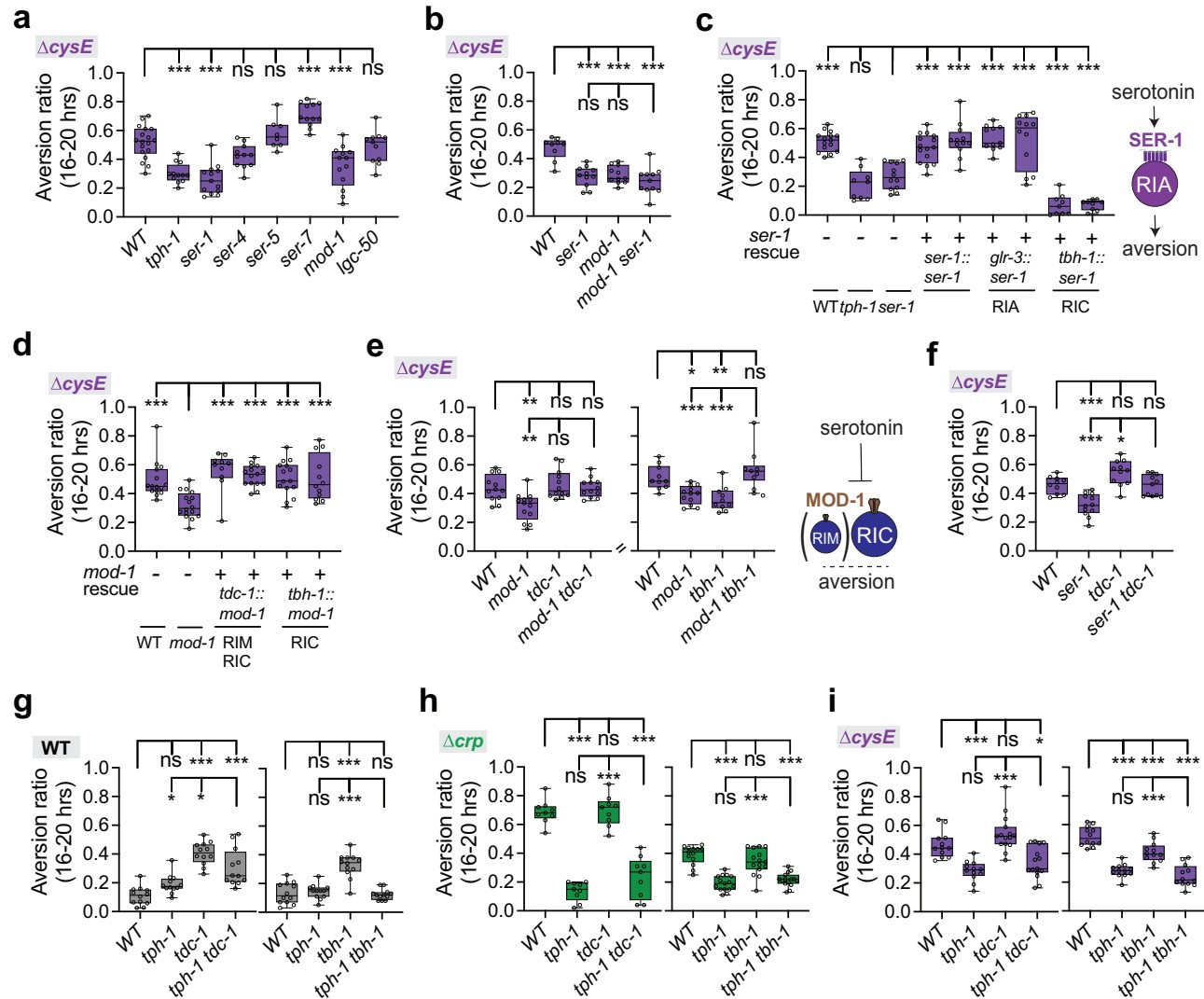

**Fig. 7 | Aversion from the mediocre ΔcysE diet requires two serotonin receptors, SER-1 and MOD-1. a** Behavioral responses of wild-type, *tph-1*, and serotonin receptor mutant animals on the mediocre ΔcysE diet. Aversion is diminished in *ser-1* and *mod-1* mutants (n = 8–18 assays). **b** Genetic interaction of *mod-1* with *ser-1* on the mediocre ΔcysE diet (n = 9–11 assays). **c** Aversion behavior of *ser-1* mutants expressing *ser-1* cDNA from its endogenous promoter (*ser-1*) or promoters for RIA (*glr-3*) or RIC neurons (*tbh-1*). Two transgenic lines were tested for each plasmid (n = 9–15 assays). **d** Aversion behavior of *mod-1* rescued in RIM and/or RIC neurons. Two transgenic lines were tested for each plasmid (n = 12–16) assays. **e** Genetic interaction of *mod-1* with *tdc-1* or *tbh-1* on the ΔcysE diet and schematic model of MOD-1 action. Stopped and dotted lines represent inhibition of RIC and

octopamine (n = 10-13 assays). **f** Genetic interaction of *ser-1* with *tdc-1* on the mediocre ΔcysE diet (n = 10-12 assays). Genetic interactions of *tph-1, tdc-1*, and *tbh-1* across bacterial diets. (**g**) wild-type diet (n = 12 assays); (**h**) Δ*crp* diet (n = 9–15 assays); (**i**) ΔcysE diet (n = 12-15 assays). **g–i** use the *tdc-1(n3420)* allele; all other figures and panels use *tdc-1(n3419)*. For (**a–f**), each data point indicates individual assay. Results are shown with median ± quartiles in boxes and Min to Max whiskers. ns, not significant, *P < 0.05, **P < 0.01, ***P < 0.001 by one-way ANOVA corrected by Dunnett's multiple comparisons (**a, c, d**) or Tukey's multiple comparisons (**b, e–i**). Full genotypes of all *C. elegans* strains are provided in Supplementary Data 3. All sample sizes, statistical tests used, and exact P values are provided in Supplementary Data 4. Source data are provided as a Source Data file.

These experiments were conducted on BW25113, an *E. coli* K12 strain that has been widely used to study metabolism in *C. elegans*, and not on the typical OP50 *E. coli* B strain. While these bacteria are approximately equal in attractiveness, they are not metabolically identical[19]. For example, although animals develop quickly and feed well on K12 strains compared to the OP50 B strains, they have reduced lipid stores[86–88] and altered lifespan regulation caused by changes in bacterial lipopolysaccharide (LPS)[89]. Conversely, OP50 is a poor source of the essential vitamin B12 and can result in slower larval development of *C. elegans*[9,42,86] and thus may engage different behavioral and stress responses than K12 bacteria.

The neurotransmitter serotonin is required for *C. elegans* to distinguish between high-quality and mediocre *E. coli* diets, with bidirectional roles in guiding behavior (Fig. 8a, b). Serotonin has long been

associated with positive aspects of food quality in *C. elegans*, and stimulates behaviors such as feeding, egg-laying, and dwelling that are observed in beneficial conditions. However, serotonin has a prominent role in behavioral and physiological responses to toxic foods in mammals[90], and in *C. elegans*, serotonin is required for the avoidance of toxic or pathogenic bacteria[14,32] and for the learned avoidance of bacterial pathogens[20]. Using cell-specific knockouts and acute silencing experiments, we found that high-quality and mediocre diets were interpreted by different classes of serotonergic neurons. The serotonergic ADF neurons supported retention on a high-quality bacterial diet, while serotonergic NSM neurons and to a lesser extent HSN neurons drove aversion from mediocre diets (Fig. 8b).

These results add to existing evidence that serotonin and individual serotonergic neurons have context-dependent functions. For

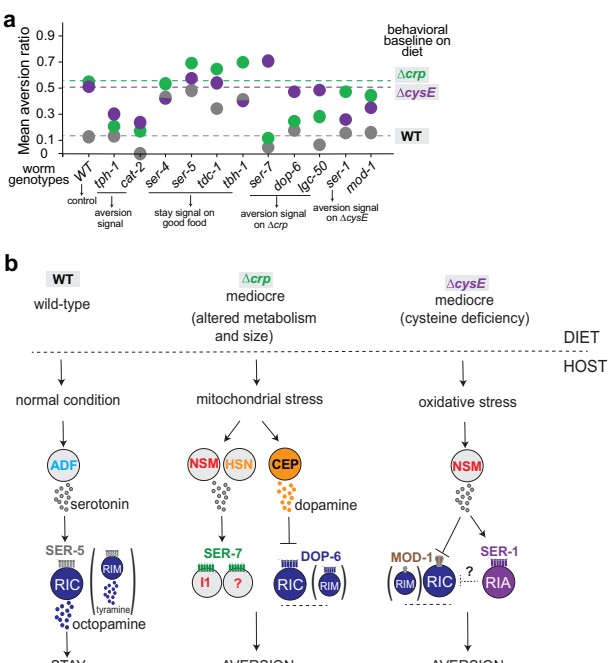

**Fig. 8 | Summary and model of how *C. elegans* distinguishes food quality.**
**a** Neurotransmitters and receptors that regulate aversion on the wild-type diet and mediocre Δ*crp* or Δ*cysE* diets. Mean aversion ratios across all assays were used to generate the bubble map; dotted lines represent the mean aversion ratio of wild-type animals on each diet; statistical comparisons appear in the main figures.
**b** Preliminary neural circuits for aversion behavior; additional neurons remain to be defined. Arrows represent activation of neurons, transmitters, or behaviors; stopped and dotted lines represent inhibition.

example, ADF serotonin drives increased feeding on familiar bacteria[21,80], possibly aligning with the appetitive retention on the (familiar) high-quality diet observed here and attraction to bacterial metabolites[33]. However, ADF serotonin can also drive learned avoidance of pathogenic bacteria, an aversive behavior[20,91]. Conversely, serotonin from NSM neurons supports dwelling behavior on food and slow locomotion upon food encounters, which are appetitive behaviors[11,31], but our results indicate that NSM serotonin can drive avoidance of mediocre foods. While this finding appears surprising, it matches prior results showing that NSM serotonin drives slow avoidance of food spiked with the mitochondrial toxin antimycin A[17]. Moreover, antimycin exposure causes NSM to change its properties such that calcium levels increase after food removal rather than food addition[17]. Together, these results argue that serotonergic neurons respond to metabolic history and not just immediate sensory inputs from food. In agreement with this hypothesis, serotonin synthesis in both ADF and NSM is regulated by food history[80,92], serotonin levels in ADF are increased by pathogen exposure[20], and serotonin synthesis in NSM is induced by mitochondrial toxins[32].

By screening for aversion behavior on three different bacterial diets, we identified roles for all six known serotonin receptors in the aversion assay. Two of the serotonin receptors, *ser-4* and *ser-5*, were required to suppress aversion from a high-quality diet; the other four receptors, *ser-7, lgc-50, mod-1,* and *ser-1,* supported aversion from two mediocre diets (Fig. 8a). Some serotonin receptors distinguished among mediocre foods, such that *ser-7* only affected aversion to Δ*crp*, whereas *ser-1* only affected aversion to Δ*cysE*. This double dissociation indicates that the nervous system can discriminate among aversive contexts. All six serotonin receptors also participate in serotonin-dependent locomotor slowing when *C. elegans* enters a bacterial

lawn[35]. Slowing on lawn entry relies on NSM and HSN serotonergic neurons, like aversion from mediocre diets, but the contributions of receptors are different: *mod-1, ser-4,* and *lgc-50* have partly redundant functions that promote slowing behavior, while *ser-1, ser-5,* and *ser-7* have modulatory roles[35]. In another assay, food-dependent regulation of nociception, the serotonin receptors *ser-1, ser-5,* and *mod-1* have congruent but non-redundant roles[93]. These comparisons highlight the distinctiveness of serotonin signaling in different contexts. We note that serotonin-deficient *tph-1* mutants move rapidly on food, but are largely normal in their locomotion[32,35,94], as are the single-cell serotonin knockouts that elicit aversion behavior[11]. The six serotonin receptor mutants, as well as dopamine and octopamine mutants, also have largely normal locomotion on food[32,35,94,95].

In addition to serotonin, the monoamine transmitters dopamine and octopamine contribute to aversion behavior. Dopamine promotes aversion from both mediocre diets, while octopamine suppressed aversion from all diets. Dopamine is also required for aversion from bacteria spiked with the mitochondrial toxin antimycin[83], but the Δ*crp* diet and antimycin engage different dopamine receptors[83]. Like serotonergic neurons, dopamine neurons can reshape their properties based on context: exposure to a low-quality diet increases dopaminergic neuron responses to bacterial conditioned medium[72].

The octopaminergic RIC neurons oppose the action of serotonin from NSM, suppressing aversion from all bacterial lawns. Octopamine and tyramine also suppress aversion from the bacterial pathogen *Pseudomonas aeruginosa* PA14, which downregulates expression of the biosynthetic enzyme *tdc-1* in RIM and RIC neurons to reduce tyramine and octopamine signaling and enhance aversion[16]. We found that the serotonin and dopamine receptors *ser-5, mod-1,* and *dop-6* could all modulate aversion when expressed in the octopaminergic RIC neurons, alone or together with the tyraminergic RIM neurons. Other serotonin targets such as RIA and I1 neurons appear selective for particular bacterial contexts, although future studies may reveal additional connections among them. We did not test all known sites of serotonin receptor expression, so the neurons identified here may not be the only ones that can affect aversion.

Serotonin and tyramine/octopamine have antagonistic functions in *C. elegans* egg-laying, feeding regulation, and nociception[74,79,80,96]. We observe similar antagonism on mediocre diets, but our results on the high-quality diet show that serotonin can potentiate RIC and octopamine signaling in certain conditions. Tracing the interactions between these neurons and transmitters will provide more insight into these neuromodulatory circuits and their responses to different bacterial diets. More generally, the parallels and contrasts between these wireless circuits for aversion provide insight into the logic of neuromodulatory integration across molecules and contexts.

## Methods
A list of Reagents and Resources used in this study is available in Supplementary Data 2.

The full genotypes of strains shown in each figure are provided in Supplementary Data 3.

### *C. elegans* strains
All *C. elegans* strains were maintained on nematode growth medium (NGM) agar plates seeded with bacterial diet *E. coli* BW25113 at 20 °C. CGC1 (formerly known as PD1074) was provided by the Caenorhabditis Genetics Center (CGC) and used as the wild-type strain. All strains used in this study are listed in Supplementary Data 2.

### Bacterial strains
*E. coli* strains OP50, DA837, HB101, BW25113, and *Comamonas aquatica* strain DA1877 were cultured in Lysogeny Broth (LB) liquid medium at 37 °C. *Pseudomonas aeruginosa* strain PA14 was cultured in Lysogeny Broth (LB) liquid medium at 25 °C. *E. coli* Keio deletion mutants[45] were

purchased from the National Bioresource Project of Japan (NBRP) and were grown at 37 °C in LB medium with 25 µg/mL kanamycin.

### Generation of transgenes

To construct plasmids for expression of *ser-5* in *C. elegans* neurons, promoter sequences of *ser-5* (2001bp), *glr-7* (2000bp), *tdc-1* (4550 bp) or *tbh-1* (4537 bp) were inserted into the *pSM::SL2::GFP::short unc-54 3' UTR* vector along with a *ser-5* cDNA. Injection mixtures containing *Pser-5::ser-5::SL2::GFP* (30 ng/uL), *Pglr-7::ser-5::SL2::GFP* (30 ng/uL), *Ptdc-1::ser-5::SL2::GFP* (30 ng/uL), or *Ptbh-1::ser-5::SL2::GFP* (30 ng/uL) together with the co-injection marker *Pmyo-3::mCherry* (3 ng/uL) were injected into CX13569 *ser-5(tm2647)* I.

To construct plasmids for expression of *ser-7* in *C. elegans* neurons, promoter sequences of *ser-7* (2118 bp), *flp-21* (2003bp), *del-7* (839 bp), *ceh-28* (1981bp), *ceh-19* (1496 bp), *ceh-53* (1476 bp), *ceh-45* (2022 bp), *lgc-8* (1999bp), or *flp-15* (2402 bp) were inserted into the *pSM::SL2::GFP::short unc-54 3' UTR* vector along with a *ser-7* cDNA. Injection mixtures containing *Pser-7::ser-7::SL2::GFP* (30 ng/uL), *Pflp-21::ser-7::SL2::GFP* (30 ng/uL), *Pdel-7::ser-7::SL2::GFP* (30 ng/uL), *Pceh-28::ser-7::SL2::GFP* (30 ng/uL), *Pceh-19::ser-7::SL2::GFP* (30 ng/uL), *Pceh-53::ser-7::SL2::GFP* (30 ng/uL), *Pceh-45::ser-7::SL2::GFP* (30 ng/uL), *Plgc-8::ser-7::SL2::GFP* (30 ng/uL), or *Pflp-15::ser-7::SL2::GFP* (30 ng/uL) together with the co-injection marker *Pmyo-3::mCherry* (3 ng/uL) were injected into DA2100 *ser-7(tm1325)* X.

To construct plasmids for expression of *ser-1* in *C. elegans* neurons, promoter sequences of *ser-1* (2103 bp), *tbh-1* (4537 bp), or *glr-3* (2838 bp) were inserted into the *pSM::SL2::GFP::short unc-54 3' UTR* vector along with a *ser-1* cDNA. Injection mixtures containing *Pser-1::ser-1::SL2::GFP* (30 ng/uL), *Ptbh-1::ser-1::SL2::GFP* (30 ng/uL), or *Pglr-3::ser-1::SL2::GFP* (30 ng/uL) together with the co-injection marker *Pmyo-3::mCherry* (3 ng/uL) were injected into DA1814 *ser-1(ok345)* X.

To construct plasmids for expression of histamine-gated chloride channels in order to chemically silence *C. elegans* neurons, promoter sequence of *srh-142* (3976 bp), or *del-7* (839 bp) were inserted into the *pSM::SL2::mCherry::short unc-54 3' UTR* vector along with the HisCl1 coding sequence. Injection mixtures containing *Psrh-142::HisCl1::SL2::mCherry* (20 ng/uL) together with the co-injection marker *Pelt-2::NLS::GFP* (2.5 ng/uL), or *Pdel-7::HisCl1::SL2::mCherry* (20 ng/uL) (no coinjection marker) were injected into PD1074.

To construct plasmids for expression of *cat-2* in *C. elegans* neurons, promoter sequences of *cat-2prom3* (1143 bp), *dat-1* (786 bp), *hlh-17* (3008 bp), or *elt-2* (5049 bp) were inserted into the *pSM::SL2::GFP:: unc-54 3' UTR* vector along with the *cat-2* genomic coding region. Injection mixtures containing *Pcat-2prom3::cat-2::SL2::GFP* (30 ng/uL), *Pdat-1::cat-2::SL2::GFP* (30 ng/uL), *Phlh-17::cat-2::SL2::GFP* (30 ng/uL), or *Pelt-2::cat-2::SL2::GFP* (30 ng/uL) together with the co-injection marker *Pmyo-3::mCherry* (5 ng/uL) were injected into CX11078 *cat-2(e1112)* II.

To construct plasmids for expression of *dop-6* in *C. elegans* neurons, promoter sequences of *tdc-1* (4550 bp) or *tbh-1* (4537 bp) were inserted into the *pSM::SL2::GFP::short unc-54 3' UTR* vector along with a *dop-6* cDNA. Injection mixtures containing *Ptdc-1::dop-6::SL2::GFP* (30 ng/uL), or *Ptbh-1::dop-6::SL2::GFP* (30 ng/uL) together with the co-injection marker *Pmyo-3::mCherry* (3 ng/uL) were injected into RB1680 *dop-6(ok2090)* X.

To construct plasmids for expression of *mod-1* in *C. elegans* neurons, promoter sequences of *tdc-1* (4550 bp) or *tbh-1* (4537 bp) were inserted into the *pSM::SL2::GFP::short unc-54 3' UTR* vector along with a *mod-1* cDNA. Injection mixtures containing *Ptdc-1::mod-1::SL2::GFP* (30 ng/uL), or *Ptbh-1::mod-1::SL2::GFP* (30 ng/uL) together with the co-injection marker *Pmyo-3::mCherry* (3 ng/uL) were injected into MT9668 *mod-1(ok103)* V.

To construct plasmids for inverted rescue of *tdc-1* in *C. elegans* neurons, the promoter sequence of *tdc-1* was assembled with the *pSM* vector together with the inverted/floxed *tdc-1* cDNA-

*sl2GFP*. Injection mixtures containing *Ptdc-1::inverted/floxed tdc-1 cDNA-sl2GFP* (60 ng/uL) together with the co-injection marker *Pmyo-3::mCherry* (3 ng/uL) were injected into MT13113 *tdc-1 (n3419)* II, to generate the inverted *tdc-1* rescue background line. This line with 'No Cre' was identified as *tdc-1; kyIs837*. *kyIs837* was a spontaneously integrated allele that was backcrossed three times. To generate rescue lines for neurons, transgenic lines containing *Primb-1::nCre* (5 ng/uL) or *Pdmsr-7::nCre* (3 ng/uL) together with the co-injection maker *myo-2::mCherry* (1 ng/uL) were crossed with the inverted *tdc-1* rescue background line.

### Screen for *C. elegans* aversion to *E. coli* Keio knockout mutants

To prepare bacteria for the primary behavioral assay, individual *E. coli* deletion mutants were grown in LB medium with 25 µg/mL kanamycin at 37 °C for about 16 h. 12.5 µL overnight cultures were seeded onto 12-well plates containing 3.5 mL standard NGM agar in each well, dried, grown at 37 °C for one day and incubated at room temperature for another day. Animals were maintained on the standard wild-type bacteria BW25113 for at least two generations before the behavioral assay. About 15-20 L4 animals were placed in each well of 12-well plates that contained BW25113_WT or individual *E. coli* mutants and incubated at 21 °C for 20 h. Plates were then visually screened for the fraction of animals that exhibited aversive behavior, defined as (animals outside the bacterial lawn)/(total number of animals). Each *E. coli* deletion mutant was screened at least twice. A secondary screen was performed for identified positive hits from the primary screen, using the same method. Positive hits identified from the secondary screen were further verified in assays on 6-well plates that were video-tracked for 20 h (recording rate 1 frame per minute, 1200 frames per video) with at least three independent replicates. Behavioral rigs were equipped with four 15 MP cameras (PL-D7715, Pixelink), one per six-well plate, and videos were cropped to individual wells and analyzed every six frames (200 data points per condition) using published MatLab codes (https://doi.org/10.5281/zenodo.10723701)[16]. Wild-type *C. elegans* on wild-type BW25113 bacteria were defined as having low aversive behavior (aversion ratio less than 0.2). Bacterial mutants that induced significantly higher aversion than BW25113_WT were chosen for subsequent studies.

### Genotyping identified *E. coli* mutants

Bacterial colonies from individual bacterial mutants were streaked on LB agar plates, grown at 37 °C, and genotyped by PCR using the kanamycin-cassette-specific primers (forward primer 5'-CGGTGCCCTGAATGAACTGC-3', reverse primer 5'-CAGTCA-TAGCCGAATAGCCT-3') and genomic primers for individual bacterial mutants, which were designed at starting 100-300 bases upstream of start-codon (forward primer) or downstream of stop codon (reverse primer). The correct fragment sizes were verified.

### Quantification of *Pdaf-7::gfp*

*Pdaf-7::gfp* animals at the L4 stage were transferred onto behavioral assay plates seeded with *E. coli* wild-type (BW25113) or test bacteria and incubated at 21° C for 16 h. Animals mounted on 2% agarose pads and 20 mM sodium azide were imaged under the same parameters (63x objective, Z-stack and exposure time) on the Zeiss Axioimager Z1. Quantification of mean GFP intensity was performed on maximum fluorescence intensity projections of ASI and ASJ cell bodies in FIJI.

### Fluorescent reporter quantification assays

All animals containing fluorescent markers (*hsp-6p::gfp*, *hsp-4p::gfp*, *gst-4p::gfp*, *clec-60p::gfp*, *irg-1p::gfp*, or *irg-5p::gfp*) were picked onto behavioral assay plates seeded with *E. coli* test bacteria at the L4 stage and incubated at 21 °C for 16 h, except animals with the stress reporter *gst-4:gfp*, which were incubated for 8 h before imaging. Animals mounted on 2% agarose pads and 20 mM sodium azide were imaged

on the Zeiss Axioimager Z1. As these reporters are broadly expressed, mean GFP fluorescence per pixel was quantified across the body using FIJI. Control experiments showed that the body sizes of *hsp-6p::gfp* and *gst-4p::gfp* animals grown on each bacterial strain were comparable.

## Bacterial colonization assay

Animals were treated under behavioral assay condition with indicated bacterial mutants for 12 h, picked into tubes with M9 buffers, washed three times, and placed on empty NGM plates containing 100 µg/mL carbenicillin for 1 hour. At least five animals were then individually picked into a new Eppendorf tube with M9 buffer and homogenized using a motorized tissue grinder (Fisher Scientific) coupled with a 1.5 mL plastic pestle (Fisher Scientific). The lysate was plated onto LB plates and incubated at 37 °C for 16 h. Bacterial colonies were counted manually. Three independent biological repeats were performed for *E. coli* wild-type or mutants.

## Pharyngeal pumping (feeding) assay

Animals were incubated as above for 12 h on plates seeded with indicated bacteria, then recorded using a LEICA MZ6 microscope coupled with cameras for 30 s and the number of pharyngeal contractions per animal was counted manually. At least six animals were recorded in each of two or three independent repeats.

## Brood size and development assay

For the brood size assay, an individual animal at the L4 stage was picked onto one NGM plate seeded with 200 µL bacteria grown under the conditions used in the behavioral assay. Animals were picked onto new plates every day and maintained at 20 °C. Brood size was determined by counting the total number of progeny across all plates for each animal. At least three independent biological replicates (6–8 animals for each repeat) were performed.

For the development assay, five synchronized adult worms were allowed to lay eggs for 3 h at 20 °C, on NGM plates with bacteria grown under the conditions used in the behavioral assay. Eggs were allowed to develop at 20 °C for 72 h, and scored for the number of progeny that had reached the L4 or adult stage as a fraction of all progeny.

## Lifespan assay

Seeded NGM plates used for lifespan assays were prepared with bacteria grown under the conditions used in the behavioral assay. About 20 synchronized L4 worms were picked onto assay plates and maintained at 25 °C. To separate parents from offspring, adult animals were picked onto freshly prepared assay plates daily until no further progeny were observed. Dead animals were identified by non-response to gentle prodding. Animals that were lost or crawled off agar were censored.

## Chemical supplementation

50% glucose (Millipore Sigma), 0.2 M cysteine, 0.2 M O-acetyl-serine and 40 mM PLP were prepared by dissolving powders in deionized water, filtered and stored at −20 °C. For behavioral analysis with pre-adding chemicals, stock solutions were supplemented into NGM agar immediately before pouring to desired final concentrations (glucose, 0.4%; Cys, 200 µM; OAS, 200 µM; PLP, 40 µM). Bacterial seeding and behavioral assay were performed as described in the section 'Screen for *C. elegans* aversion to *E. coli* Keio knockout mutants'. For behavioral analysis with post-adding chemicals, assay plates were prepared by first seeding 12.5 µL indicated bacterial cultures onto NGM agar for growth, as described in the section 'Screen for *C. elegans* aversion to *E. coli* Keio knockout mutants'. One hour before the behavioral assay, chemicals were directly added onto the bacterial spots to reach the final concentration under a total volume of 6 mL NGM agar (Cys, 50 µM; OAS, 200 µM; PLP, 40 µM). NGM plates without any added chemicals were assayed in parallel.

## Bacterial density experiments

Overnight bacterial cultures were grown in LB medium at 37 °C and seeded onto behavioral assay plates (Supplementary Fig. 3). Standard NGM agar plates contained 0.3% NaCl, 0.25% peptone, 25 mM potassium phosphate, 1 mM $CaCl_2$, 1 mM $MgSO_4$, 5 µg/mL cholesterol, 2.2% agar. To increase bacterial density of BW25113_WT, Δ*crp* or Δ*cysE* bacteria, 12.5 µL of overnight bacterial cultures (1x) or different concentrated suspensions of those cultures (10x, 20x, or 40x) were seeded onto NGM agar and allowed to grow for two days as indicated before behavioral tests. To decrease the bacterial density of BW25113 (WT), the behavioral assay plates contained NGM agar with low amounts of peptone (0.125%, 0.05%, or 0.025%) and were seeded with 12.5 µL of overnight bacterial culture as above. For pre-adding conditions, 200 µM cysteine or OAS were added to NGM agar when the plates were made, prior to bacterial seeding. For post-adding cysteine or OAS, assay plates were prepared by first seeding 12.5 µL of the indicated bacterial cultures onto NGM agar for two days' growth, and then cysteine or OAS was administered to the final 50 µM or 200 µM concentration in a total volume of 6 mL NGM agar, respectively. Matched pairs of agar plates were used for behavioral testing (with animals) or for determining bacterial density. Bacteria were harvested at the beginning of the behavioral assay, except for supplementation experiments (Fig. 2e and Supplementary Fig. 3i and j), in which bacteria were counted at the end of the behavioral assay. Bacteria were washed off the agar with 2 mL M9 buffer, subjected to $OD_{600}$ measurement using a DS-11 FX+ spectrophotometer (DeNovix), and bacterial density determined by serial dilution, plating on agar plates, and counting colony-forming units (CFU)[64]. We did not observe any significant differences in CFU numbers after plating on LB or NGM agar plates.

## Neuron silencing by histamine assay

1 M histamine dihydrochloride (Millipore Sigma) was prepared by dissolving powder in deionized water, filtered and stored at -20 °C. Histamine stock solution was added to NGM agar at a final concentration of 10 mM prior to seeding with bacteria. For behavioral analysis, NGM plates with or without histamine and animals with or without expression of HisCl1 channels were assayed in parallel.

## Scanning electron microscopy

Bacteria were gently eluted from assay plates and fixed in 2% glutaraldehyde, 4% formaldehyde in 0.1 M sodium cacodylate buffer, pH 7.2, for more than 1 hour at room temperature followed by overnight fixation at 4 °C. After post-fixation with 1% osmium tetroxide in 0.1 M sodium cacodylate buffer, pH 7.2 for 1 h at room temperature, samples were dehydrated in a graded series of ethanol followed by hexamethyldisilazane. Bacteria were spread onto glass coverslips and let air-dry overnight. The next morning, samples were sputter-coated with 10 nm of iridium using a Leica ACE600 sputter coater. Images were acquired in a scanning electron microscope JEOL JSM-IT500HR at 5.0 kV (JEOL USA, Inc).

## Statistical analysis

Statistical analyses were performed using GraphPad Prism and are detailed in the figure legends. Detailed statistical summaries are provided in Supplementary Data 4.

## Reporting summary

Further information on research design is available in the Nature Portfolio Reporting Summary linked to this article.

# Data availability

All data generated in this study are provided in this paper, Supplementary Information, Supplementary Data, and Source Data file. Source data are provided with this paper. All information and requests

for reagents or strains should be directed to the lead contact Cornelia Bargmann (cori@rockefeller.edu). Source data are provided with this paper.

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

## Acknowledgements

We thank Steven Flavell, Cassi Estrem, Yun Zhang, Chun-Liang Pan, Luciano Marraffini, Dennis Kim, and members of the Bargmann lab for their comments on the manuscript. We thank Robert Horvitz, William Schafer, Sreekanth Chalasani, and the Caenorhabditis Genetics Center (P40 OD010440) for sharing strains. We thank Dr. Hilda Amalia Pasolli at the Rockefeller University Electron Microscopy Facility (RRID:SCR_017792) for conducting electron microscopy. L.F. was supported by a postdoctoral fellowship from the Kavli Institute for Neural Systems. A.H. was supported by an NIH Predoctoral Fellowship. Y.G. was supported by an NSF Predoctoral Fellowship. This work was supported by a grant from the Chan Zuckerberg Initiative to C.I.B.

## Author contributions

L.F. and C.I.B. designed experiments. L.F. conducted the screen, behavioral, genetic and molecular experiments. J. M-S. and L.Y. conducted behavioral experiments and generated transgenic strains. A.H. generated and validated *tph-1* knockout strains. Y.G. generated NSM::HisCl1 transgenic strains. L.F. and C.I.B. analyzed and interpreted results and wrote the paper with input from all authors.

## Competing interests

The authors declare no competing interests.
