## [Transparent Peer Review file · Nature Communications]

Context-dependent serotonin signaling links dietary quality to foraging decisions

Corresponding Author: Professor Cornelia Bargmann

Version 0:

Reviewer comments:

Reviewer #1

(Remarks to the Author)

Overall, this is a very interesting and compelling study. The strengths of the paper include:

New Biological insights:

By leveraging the experimental accessibility of *C. elegans* and *E. coli*, the authors dissected a monoamine neural modulatory network in *C. elegans* that regulates behavioral responses to low-quality food. Using the bacterial mutants, the authors probed the signaling mechanisms that worms use to distinguish food sources of different qualities. This experimental design allows the authors to characterize the intricate neuromodulatory signaling network that responds to versatile food-generated cues to produce adaptive behaviors. The authors showed that by engaging different combinations of monoamine receptors and the activities of the neurons expressing them, this network could produce a large number of configurations to manage the dynamic environmental/food cues. These biological insights are likely to be generally applicable.

Clarity of the analysis:

Response to food is universally important and at the same time highly complex because it involves multiple regulatory signals and critically impacts various aspects of animal physiology and behavior. Meanwhile, the food itself, such as *E. coli* cells, is also complex due to its dynamic metabolism and responses to environmental conditions. Thus, it is often challenging to clearly address the question. Using the “dual model” of *C. elegans* and *E. coli*, the authors dissected the complex molecular and cellular pathways that underlie behavioral responses to different qualities of food at an impressive level of clarity.

Completeness of the analysis:

Similar to vertebrate nervous systems, serotonin acts on several receptors which have demonstrated different functions via their specific signaling mechanisms and the activities of the expressing neurons. In *C. elegans* almost half of the nervous system expresses 1 or more serotonin receptors, which also highlights the important and inevitably complicated signaling events involving serotonin. Therefore, while the critical role of serotonin in regulating a wide range of physiological and behavioral responses to food is known, a complete and mechanistic picture is largely missing. Using *E. coli* mutants with reduced food qualities, the authors characterized the function of all serotonin receptors and their interaction with dopamine and octopamine in regulating aversion of poor-quality food.

I have only a few minor suggestions.

1. In Discussion, authors described the definition of “mediocre” food. It will be informative to discuss what makes several *E. coli* mutants as “mediocre” food sources when the term first appeared in line 159.
2. The mutant *deltapdxJ* did not seem to induce *daf-7* expression or generate mitochondria UPR or oxidative stress. Feeding on this *E. coli* mutant reduces brood size. What do authors speculate about the mechanisms underlying the aversive properties of this *E. coli* mutants?
3. Does each trace in the middle panels of Fig1 represent one sample assay or the average of the assays on one mutant?

Reviewer #2

(Remarks to the Author)

The manuscript by Feng et al. describes the study of the molecular determinants of bacterial food that contribute to the attraction/aversion of *C. elegans* to a lawn of bacterial food, combining the systematic bacterial genetic analysis with the detailed *C. elegans* host genetic characterization of how neuron-specific activities of serotonin production and signaling can act on specific sets of serotonin receptors to mediate responses to bacteria.

There have been a number of studies demonstrating how pathogenic bacteria or other bacteria that induce *C. elegans* stress can cause aversion of the bacterial lawn. Feng et al. have begun to unravel the molecular mechanisms at play in the bacterial lawn. The authors define 22 different mutants that can elicit aversive behavior. Interestingly, over 200 mutants that do not support wild-type levels of growth do not induce lawn avoidance, suggesting that the aversive response is not simply a readout of "bad" (i.e. suboptimal nutritive) food. This study, conducted through an elegant unbiased approach, illustrates the complexity of how changes in bacterial metabolism can have dramatic effects on the host.

To infer what is going on in the host, the authors begin by systematically examine the induction of different stress sensors in *C. elegans*, identifying distinct and not necessarily overlapping induction of stress sensors by different bacterial mutants able to elicit avoidance behavior, suggesting that no single type of stress is necessarily required, but in fact lawn avoidance may reflect the balance between attractive and repulsive forces, to which many processes contribute. The authors focus on characterization of two bacterial mutants that induce mitochondrial and oxidative stress to dissect the host response circuitry involved. The authors examine the role of serotonin signaling through the selective neuron-specific deletion of *tph-1* from each of ADF, NSM, and HSN neurons, revealing strikingly distinct neuron-specific roles for TPH-1. The authors identify a role for *tph-1* expression from the ADF neurons in mediating attraction to wild-type bacteria, and a role for *tph-1* expression in the NSM and HSN neurons in mediating repulsion to "mediocre" or stress-inducing bacterial mutants. The authors also pinpoint a role for each of the known serotonin receptors in mediating these distinct activities of serotonin signaling.

The work is nicely conceived and rigorously executed, and an important contribution to the understanding of how microbial metabolism and physiology can influence the nervous system signaling of animal hosts. Also revealed are data pertaining to how specific neuronal types in neurotransmitter signaling can signal in balancing ways to regulate behavioral responses to sensory cues. As such, the work should be of considerable interest to investigators working in *C. elegans* host-microbe interactions as well as the broader community of investigators working to understand the effects of the microbiota on host animal physiology.

I have only a single comment for revision regarding placing the current work in a clearer context relative to prior work, much of it from the authors' group that has pioneered this field. Doing so may clarify some of the nuances underlying this field for a more general audience. Serotonin signaling has been previously implicated in *C. elegans* behavioral responses to bacteria. Enhanced slowing in response to an initial encounter between food-deprived *C. elegans* and a bacterial lawn was previously shown to require serotonin (Sawin, 2000). Zhang et al. (2005) showed that *tph-1* expression was upregulated in the ADF neurons in response to exposure to pathogenic *Pseudomonas aeruginosa*, and *tph-1* activity in the ADF neurons was required for learned aversion of *P. aeruginosa* in concert with MOD-1. Flavell et al. (2013) showed that *tph-1* expression in the NSM and HSN neurons promotes dwelling behavior on nutritive bacteria by acting on MOD-1.

Ostensibly, the current study seems to arrive at conclusions that are conceptually the opposite of the 2005 and 2013 studies. That is, Zhang et al. suggests that TPH-1 in ADF mediates aversion to stressful bacteria, and Flavell et al. suggests that TPH-1 in NSM and HSN mediates attraction to nutritious bacterial food.

I don't think there is an experimental contradiction here, but some clarification may be helpful. Lawn-leaving behavior, choice/preference behavior, and roaming/dwelling foraging behavior are all different assays. Lawn-leaving is perhaps the least-specific, but at the same time, represents perhaps the "bottom line" decision of the animal regarding a bacterial food source. Multiple factors contribute to the lawn-leaving phenotype in ways that may counteract one another. A more precise explanation of the differences in the assays and what factors may be involved, and specific explanation of the context in which the terms preference, aversion, and foraging are being used may be helpful to the more general audience to better appreciate the important take-home messages of the work.

Reviewer #3

(Remarks to the Author)

Review of 'Context-dependent serotonin signaling links dietary quality to foraging decisions'

Summary of the paper:

The manuscript by Feng et al., describes a serotonergic circuit in the nematode *C. elegans* that appears to be able to define mediocre quality and thus aversive foods. The study makes use of a library bank of bacterial knock-out strains to ask an interesting question: What aspects of food signaling modulates foraging decisions?

Overall, I found the study to be interesting, but my enthusiasm is tempered by certain experimental choices that leave the conclusions not entirely supported.

First, the authors use a behavioral assay on a library of mutant *E. coli* strains to measure the aversion ratio, the number of animals on versus off the food spot. First, they show that this captures the response to a *P. aeruginosa* strain which is pathogenic. They then move on to screen for multiple *E. coli* mutants and identify a number of relevant metabolic pathways

that increase worm aversion to these bacterial foods. They then move on to the circuits in the animal responsible for this aversion and find neuromodulators and their receptors involved in aversion, specifically serotonin, but they also find a minor role for octopamine/tyramine.

Overall, the screen and the behavioral assay are interesting, however the presentation and interpretation of the results raised some concerns I detail below. Overall, I found that due to the broad nature of this study, the individual results reads somewhat unfocused at times, and not well embedded in the rich literature on neuromodulators and specifically serotonin pathways in *C. elegans* foraging. In summary, I found that despite the intriguing title the manuscript fails to convincingly demonstrate that serotonergic signaling is responsible to encode dietary quality.

I have a number of major concerns with this study:

While the simplified assay is a great tool for rapid screening, it can be design only give a coarse view of aversion, which could be due to a number of factors (satiety, pathogenicity, nutritional balance, ingestibility of the food). For example, the observation that animals on HB101 have a higher aversion rate than other *E. coli* strains (Fig. 1b) seems unusual as HB101 is canonically used for hard-to-grow strains or eat-mutants, as it better supports growth. This suggests that satiety is potentially confounded with aversion, which could also be the case in the screen.

Multiple studies have investigated the density dependence and food choice in *C. elegans*. A recent paper studies spot leaving in a multiple patch arena of the same bacteria or different bacteria at different densities (Madirolas et al., 2023) and found that to first order the preference is entirely determined by bacterial density. This is the basis for a second potential confound in this study: The authors study multiple *E. coli* mutants, most of which act on bacterial metabolism, which very likely impacts growth rate and culture density, and possibly impacts cell size. At minimum, a control for cell density and CFU counts should be presented to see if density is a concern, beyond just OD600 before spotting on the plate. In addition, as bacterial were allowed to grow on NGM agar (a minimal medium, but nonetheless bacteria will grow), which could further exacerbate density differences. This confound could also explain the fact that glucose supplementation suppressed aversion to the *crp* mutant.

Fluorescence reporters of stress pathways:

A potential concern for the induction of the stress genes is the observed developmental and feeding difference between the bacterial strains: It would be better to normalize to either animal size and/or a stable marker to ensure that the size is not a concern or a driver of the observed effects.

The authors then move on to investigate which pathways in the worm are responsible for food aversion. They present data for the major neuromodulators. Unfortunately, only one allele is shown per neuromodulator, and at least for 5-HT the two alleles have quite some behavioral differences, specifically in the severity of the feeding and locomotion defects. It would be good to confirm the major results with a second allele.

The expression of cDNA of *ser-5* in pharyngeal neurons is unexpected - why did the authors assume *ser-5* to be relevant in pharyngeal neurons? Based on single-cell RNAseq, *ser-5* is only in a single pharyngeal neuron class (l3) that isn't implicated in being important for feeding. Instead, *ser-7* is the relevant 5-HT receptor in the pharynx (as seen in a later figure). How do the authors envision pharyngeal *ser-7* acts on a locomotion phenotype 'aversion'?

The feeding rate measurements on the selected strains are a bit surprising: Previous studies suggested that pumping is reduced on 'mediocre' foods (Shtonda and Avery, 2006).

The size difference in bacteria is interesting, and not unexpected given the specific pathways affected. However, it also adds another confound as it has been shown that ingestion is size-selective and it doesn't seem to match with the pumping rate data, which would be expected to be lower for larger bacteria. Taken together with the complex confound of satiety and density, I am unsure if the results are easily interpretable as stated.

Overall, the results section would be improved by providing a rationale for certain experiments e.g., specific rescues beyond stating significance or non-significance. I would therefore also recommend overall editing and stream-lining the presentation to clarify the specific findings and to contextualize these within the literature.

Minor concerns:

The description of the bacterial pathways in the results section is very lengthy and could be edited to be more concise while retaining the relevant information.

Allele names are omitted in the main text. It would be nice to include these in the manuscript for easier comparability to the many other studies that have also used the main neuromodulator mutants.

Version 1:

Reviewer comments:

Reviewer #1

(Remarks to the Author)

The authors addressed all my questions and comments in their revised manuscript.

Reviewer #2

(Remarks to the Author)

The reviewers have satisfactorily addressed my comments with a concise, insightful perspective on the complex activities of the ADF neurons and serotonin signaling in mediating responses to bacteria.

Reviewer #3

(Remarks to the Author)

The authors have substantially revised the text to improve the clarity of the reporting. In addition, the density experiments added a new angle to the interpretation of the results, and the additional alleles improve the robustness of the conclusions. The discussion of their interesting results - that modulatory pathways seem to drive apparently opposite behaviors - is great. It helps to contextualize their findings and now cites a large body of relevant literature.

The improved display of results figures, as well as the textual changes make the study clear and impactful. I recommend publication and appreciate the authors efforts in responding to my and the other reviewers comments.

Summary of revisions:

We appreciate the thoughtful and constructive comments from the reviewers. We have revised the manuscript with new data and discussion to address their concerns. Major changes in the main text and supplementary information are highlighted in red.

First, we added a new analysis of bacterial density for all strains, both by OD₆₀₀ and by counting colony-forming units, to allow us to relate cell number to behavioral responses. By varying cell densities across the bacterial strains in the assay, we show that bacterial cell number does not explain behavioral aversion (**Fig. 2e and Supplementary Fig. 3**).

Second, we added data for a second allele of *tph-1* (serotonin synthesis) (**Supplementary Fig. 4a**), and clarified use of a second allele of *tdc-1* (tyramine/octopamine synthesis) (Figure 7).

Third, we added a new analysis of feeding behavior for wild-type animals on different bacteria (**Supplementary Fig. 2**), as well as feeding behavior of the serotonin receptor mutant *ser-7* (**Supplementary Fig. 4b**).

Fourth, we clarified the text and placed the work in the context of other work in the field. We emphasize the relationship of the aversion behavior to stress responses (pathogens, toxins, and inedible food) and link our results to previous studies of those responses.

Point-by-point responses are described below, with citations of figure and line numbers.

Reviewer #1 (Remarks to the Author):

Overall, this is a very interesting and compelling study. The strengths of the paper include:

New Biological insights:

*By leveraging the experimental accessibility of *C. elegans* and *E. coli*, the authors dissected a monoamine neural modulatory network in *C. elegans* that regulates behavioral responses to low-quality food. Using the bacterial mutants, the authors probed the signaling mechanisms that worms use to distinguish food sources of different qualities. This experimental design allows the authors to characterize the intricate neuromodulatory signaling network that responds to versatile food-generated cues to produce adaptive behaviors. The authors showed that by engaging different combinations of monoamine receptors and the activities of the neurons expressing them, this network could produce a large number of configurations to manage the dynamic environmental/food cues. These biological insights are likely to be generally applicable.*

Clarity of the analysis:

*Response to food is universally important and at the same time highly complex because it involves multiple regulatory signals and critically impacts various aspects of animal physiology and behavior. Meanwhile, the food itself, such as *E. coli* cells, is also complex due to its dynamic metabolism and responses to environmental conditions. Thus, it is often challenging to clearly address the question. Using the “dual model” of *C. elegans* and *E. coli*, the authors dissected the*

complex molecular and cellular pathways that underlie behavioral responses to different qualities of food at an impressive level of clarity.

Completeness of the analysis:

*Similar to vertebrate nervous systems, serotonin acts on several receptors which have demonstrated different functions via their specific signaling mechanisms and the activities of the expressing neurons. In *C. elegans* almost half of the nervous system expresses 1 or more serotonin receptors, which also highlights the important and inevitably complicated signaling events involving serotonin. Therefore, while the critical role of serotonin in regulating a wide range of physiological and behavioral responses to food is known, a complete and mechanistic picture is largely missing. Using *E. coli* mutants with reduced good qualities, the authors characterized the function of all serotonin receptors and their interaction with dopamine and octopamine in regulating aversion of poor-quality food.*

I have only a few minor suggestions.

*1. In Discussion, authors described the definition of “mediocre” food. It will be informative to discuss what makes several *E. coli* mutants as “mediocre” food sources when the term first appeared in line 159.*

We added a rationale for the use of the term “mediocre” when we summarize the effects of the bacterial mutants (lines 201-202).

*2. The mutant $\Delta pdxJ$ did not seem to induce *daf-7* expression or generate mitochondria UPR or oxidative stress. Feeding on this *E. coli* mutant reduces brood size. What do authors speculate about the mechanisms underlying the aversive properties of this *E. coli* mutants?*

The $\Delta pdxJ$ mutant probably affects both *C. elegans* and *E. coli*. Vitamin B6 is an essential nutrient for *C. elegans* grown in axenic media (Zečić *et al.*, 2019, Genes Nutr), and postembryonic *C. elegans* growth on a poor-quality bacterial diet requires the same *E. coli* vitamin B6 biosynthetic genes that we identified here, including *pdxJ* (Feng *et al.*, 2024, Comm Biol). Vitamin B6 is an essential cofactor in multiple biochemical pathways that could affect *C. elegans* metabolism, growth, and behavior, including serotonin and dopamine synthesis. In addition, adding vitamin B6 (PLP) to the $\Delta pdxJ$ strain immediately before the behavioral assay did not rescue aversion, suggesting that the mutation leads to loss of a second bacterial factor. Among the many bacterial vitamin B6-dependent proteins, some overlap with other “mediocre” pathways identified here: *cysK* in the cysteine biosynthesis cassette and *rffA* (*wecE*) in the ECA biosynthesis pathway (Tramonti *et al.*, 2021, EcoSalPlus). We added a summary of these points to lines 136-142 and 185-188.

3. Does each trace in the middle panels of Fig1 represent one sample assay or the average of the assays on one mutant?

Each trace represents the average of 9-15 biological replicates. To improve clarity, we have updated the figure with the number of replicates and added a shaded region representing the 95% confidence interval.

Reviewer #2 (Remarks to the Author):

The manuscript by Feng et al. describes the study of the molecular determinants of bacterial food that contribute to the attraction/aversion of C. elegans to a lawn of bacterial food, combining the systematic bacterial genetic analysis with the detailed C. elegans host genetic characterization of how neuron-specific activities of serotonin production and signaling can act on specific sets of serotonin receptors to mediate responses to bacteria.

There have been a number of studies demonstrating how pathogenic bacteria or other bacteria that induce C. elegans stress can cause aversion of the bacterial lawn. Feng et al. have begun to unravel the molecular mechanisms at play in the bacterial lawn. The authors define 22 different mutants that can elicit aversive behavior. Interestingly, over 200 mutants that do not support wild-type levels of growth do not induce lawn avoidance, suggesting that the aversive response is not simply a readout of “bad” (i.e. suboptimal nutritive) food. This study, conducted through an elegant unbiased approach, illustrates the complexity of how changes in bacterial metabolism can have dramatic effects on the host.

To infer what is going on in the host, the authors begin by systematically examine the induction of different stress sensors in C. elegans, identifying distinct and not necessarily overlapping induction of stress sensors by different bacterial mutants able to elicit avoidance behavior, suggesting that no single type of stress is necessarily required, but in fact lawn avoidance may reflect the balance between attractive and repulsive forces, to which many processes contribute. The authors focus on characterization of two bacterial mutants that induce mitochondrial and oxidative stress to dissect the host response circuitry involved. The authors examine the role of serotonin signaling through the selective neuron-specific deletion of tph-1 from each of ADF, NSM, and HSN neurons, revealing strikingly distinct neuron-specific roles for TPH-1. The authors identify a role for tph-1 expression from the ADF neurons in mediating attraction to wild-type bacteria, and a role for tph-1 expression in the NSM and HSN neurons in mediating repulsion to “mediocre” or stress-inducing bacterial mutants. The authors also pinpoint a role for each of the known serotonin receptors in mediating these distinct activities of serotonin signaling.

The work is nicely conceived and rigorously executed, and an important contribution to the understanding of how microbial metabolism and physiology can influence the nervous system signaling of animal hosts. Also revealed are data pertaining to how specific neuronal types in neurotransmitter signaling can signal in balancing ways to regulate behavioral responses to sensory cues. As such, the work should be of considerable interest to investigators working in C. elegans host-microbe interactions as well as the broader community of investigators working to understand the effects of the microbiota on host animal physiology.

I have only a single comment for revision regarding placing the current work in a clearer context relative to prior work, much of it from the authors' group that has pioneered this field. Doing so may clarify some of the nuances underlying this field for a more general audience. Serotonin signaling has been previously implicated in C. elegans behavioral responses to bacteria. Enhanced slowing in response to an initial encounter between food-deprived C. elegans and a bacterial lawn was previously shown to require serotonin (Sawin, 2000). Zhang et

al. (2005) showed that tph-1 expression was upregulated in the ADF neurons in response to exposure to pathogenic Pseudomonas aeruginosa, and tph-1 activity in the ADF neurons was required for learned aversion of P. aeruginosa in concert with MOD-1. Flavell et al. (2013) showed that tph-1 expression in the NSM and HSN neurons promotes dwelling behavior on nutritive bacteria by acting on MOD-1.

Ostensibly, the current study seems to arrive at conclusions that are conceptually the opposite of the 2005 and 2013 studies. That is, Zhang et al. suggests that TPH-1 in ADF mediates aversion to stressful bacteria, and Flavell et al. suggests that TPH-1 in NSM and HSN mediates attraction to nutritious bacterial food.

I don't think there is an experimental contradiction here, but some clarification may be helpful. Lawn-leaving behavior, choice/preference behavior, and roaming/dwelling foraging behavior are all different assays. Lawn-leaving is perhaps the least-specific, but at the same time, represents perhaps the "bottom line" decision of the animal regarding a bacterial food source. Multiple factors contribute to the lawn-leaving phenotype in ways that may counteract one another. A more precise explanation of the differences in the assays and what factors may be involved, and specific explanation of the context in which the terms preference, aversion, and foraging are being used may be helpful to the more general audience to better appreciate the important take-home messages of the work.

The reviewer is right to ask for further explanation. Both appetitive and aversive effects of ADF serotonin and NSM serotonin are described in the existing literature, suggesting context-dependent functions. We showed that ADF serotonin drives aversion to pathogen odors (Zhang *et al.*, 2005, Nature), but Leon Avery's lab showed that ADF serotonin can mediate appetitive feeding on familiar food (Song *et al.*, 2013, eLife) and Zheng-Xing Wu's lab showed that ADF serotonin can directly stimulate feeding (Liu *et al.*, 2019, PNAS).

We and others showed that NSM serotonin drives positive food responses like dwelling (Flavell *et al.*, 2013, Cell), but Chun-Liang Pan's lab showed that NSM serotonin is upregulated by a mitochondrial toxin and then drives aversive behavior (both in the leaving assay used here, and an odor-preference behavior). Pan's lab also showed that NSM calcium responses to bacteria are remodeled by the mitochondrial toxin (Chiang *et al.*, 2022, PNAS; Tsai *et al.*, 2024, Cell Rep). There are many parallels between Pan's results with mitochondrial toxins and our results with Δcrp , which we highlight in the revised paper.

We have rewritten the discussion to make these precedents and points clearer and highlight the key conclusions (lines 402-416).

Reviewer #3 (Remarks to the Author):

Review of 'Context-dependent serotonin signaling links dietary quality to foraging decisions'

Summary of the paper:

The manuscript by Feng et al., describes a serotonergic circuit in the nematode C. elegans that appears to be able to define mediocre quality and thus aversive foods. The study makes use of a library bank of bacterial knock-out strains to ask an interesting question: What aspects of food signaling modulates foraging decisions?

Overall, I found the study to be interesting, but my enthusiasm is tempered by certain experimental choices that leave the conclusions not entirely supported.

First, the authors use a behavioral assay on a library of mutant E. coli strains to measure the aversion ratio, the number of animals on versus off the food spot. First, they show that this captures the response to a P. aeruginosa strain which is pathogenic. They then move on to screen for multiple E. coli mutants and identify a number of relevant metabolic pathways that increase worm aversion to these bacterial foods. They then move on to the circuits in the animal responsible for this aversion and find neuromodulators and their receptors involved in aversion, specifically serotonin, but they also find a minor role for octopamine/tyramine.

Overall, the screen and the behavioral assay are interesting, however the presentation and interpretation of the results raised some concerns I detail below. Overall, I found that due to the broad nature of this study, the individual results reads somewhat unfocused at times, and not well embedded in the rich literature on neuromodulators and specifically serotonin pathways in C. elegans foraging. In summary, I found that despite the intriguing title the manuscript fails to convincingly demonstrate that serotonergic signaling is responsible to encode dietary quality.

In the revised manuscript we provide a broader discussion of the roles of neuromodulators in foraging. As noted in the response to reviewer #2, we describe prior results showing that ADF and NSM serotonergic neurons have both appetitive and aversive functions (lines 402-416). We also now cite precedents for the neuromodulators we study from previous studies of stress-induced aversion. For example, serotonin from NSM neurons drives aversion to bacteria spiked with mitochondrial toxins (Chiang et al., 2022, PNAS; Tsai et al., 2024, Cell Rep), as we see for Δcrp and $\Delta cysE$. On low-quality diets, dopamine neurons are activated by food odors, and then signal to RIC via *dop-6* – as we see for aversion to Δcrp (Zhang et al., 2021, Nature Aging); dopamine has an analogous role in the avoidance of food spiked with mitochondrial toxins (Chou et al., 2022, Neurosci Res). (lines 435-441)

Antagonism between serotonin and octopamine/tyramine signaling is a common motif in *C. elegans* behavior that we did not highlight in the earlier draft. We now cite relevant references Horvitz et al., 1982, Science; Li et al., 2012, Nat Comm; Guo et al., 2018, Cell Rep; Liu et al., 2019, PNAS (lines 442-459).

I have a number of major concerns with this study:

1. While the simplified assay is a great tool for rapid screening, it can be design only give a coarse view of aversion, which could be due to a number of factors (satiety, pathogenicity, nutritional balance, ingestibility of the food). For example, the observation that animals on HB101 have a higher aversion rate than other E. coli strains (Fig. 1b) seems unusual as HB101 is canonically used for hard-to-grow strains or eat-mutants, as it better supports growth. This suggests that satiety is potentially confounded with aversion, which could also be the case in the

screen.

We agree that multiple factors can influence foraging behaviors. Specifically, we agree that long-term bacterial aversion can be caused by pathogenicity, metabolic toxicity, and un-ingestible food, and these results are supported by an extensive literature showing that toxins and pathogens trigger this behavior. Our hypothesis for the behaviors we describe here is that the bacterial foods that induce aversion cause metabolic stress ($\Delta cysE$, $\Delta fepB$, Δcrp), and are also poorly ingestible (Δcrp , $\Delta wzxE$). We make those points more clearly in the revised paper (see for example (lines 49-53, 78-83) in the introduction, (lines 357-358) in the discussion).

In the revised manuscript we emphasize the role of stress in aversion behavior, and the similarity of our results to previous studies of stress-related aversion, eg the role of serotonin from NSM. We note that Δcrp is supported by the induction of stress markers that are also induced by the established metabolic stressor $\Delta fepB$ (Fig. 2b; Zhang et al., 2019, Cell Host & Microbe). Part of the Δcrp and $\Delta wzxE$ stress is likely to be due to the large size/inedibility of the food, consistent with the induction of $daf-7$ in ASJ, which is known to result from inhibition of feeding (Boor et al., 2024, eLife). $\Delta cysE$ induces a different stress marker associated with oxidative stress.

The evaluation of dietary quality has been studied extensively in *C. elegans*, and the reviewer alludes to that work above. We think that we can exclude the alternatives that the reviewer mentions. Apologies for the length of this response, because we are not sure whether we are answering the right questions.

Satiety -- We are not sure what the reviewer is asking about satiety, but hope that we address it in our response to the question about bacterial density below (comment 2). Under the conditions we use (high food density, low animal density) satiety can lead to bouts of quiescence on the bacterial lawn but not aversion, so we do not think they can be confounded (eg You et al., 2008, Cell Metabolism; Gallagher et al., 2013, J Neurosci). Animals move freely on the lawn throughout the assays in our video recordings; if anything, they move more quickly on mediocre food, inconsistent with satiety. In addition, the genetic regulation of satiety quiescence is different from what we see here.

Bacterial strains -- The reviewer mentions HB101, and in the revised manuscript we say more about the strains used here. The standard *C. elegans* bacterial food is the *E. coli* B strain OP50, but numerous experiments have been conducted on HB101, an *E. coli* K12/B hybrid, and on the *E. coli* K12 strains BW25113/MG1655 and HT115. In this paper, we exclusively use the K12 strain background, which has also been used for many studies of metabolism and most RNAi studies in *C. elegans*. There is an extensive literature on the effect of *E. coli* diet on *C. elegans* metabolism and behavior. A reasonable summary of the field is that the K12 and B strains are metabolically different, but not that one is better than the other. For example, the K12 strain results in reduced *C. elegans* fat storage compared to the OP50 B strain and appears to provide less of the satiety factor oleic acid (Brooks and Watts, 2009, PLoS One). (The same is true for HB101, but we have not examined HB101 beyond the data in Fig. 1b, where it seems similar to the K12 strain BW25113). On the other hand, the OP50 B strain is worse source of vitamin B12, an essential nutrient (MacNeil and Walhout, 2013, Cell), and can also result in slower development, perhaps because it is harder to eat (Shtonda and Avery, 2006, J Exp Biol, but see

below). There are other differences; the *E. coli* B strain is more behaviorally attractive to *C. elegans* than it is expected to be based on growth properties (Madirolas *et al.*, 2023, Commun Biol); *C. elegans* grown on K12 and B strains have different sensitivity to the lifespan-regulating gene *nmur-1* (Maier *et al.*, 2010, PLoS Biol); and so on. *E. coli* is not thought to be a natural food source for *C. elegans*, so all of these experiments are related to lab conditions.

At a sensory level, *E. coli* B and K12 strains are sensed as “different” by *C. elegans*: animals trained with *E. coli* B and a toxin subsequently avoid B odors in a choice with K12 odors, and conversely animals trained with *E. coli* K12 and a toxin subsequently avoid K12 odors in a choice with B odors (Chiang *et al.*, 2022, PNAS). At baseline the strains are approximately equal in attractiveness.

We note that the OP50 *E. coli* B strain used in most labs is distinct from the DA837 *E. coli* B strain, a derivative of OP50. DA837 is particularly hard for young larvae to eat (Brooks and Watts, 2009, PLoS One) and is flagged as a poor-quality food in the *Caenorhabditis* strain collection (<https://cgc.umn.edu/strain/DA837>). DA837 is the strain that shows the largest differences from HB101 in various settings of growth or feeding; the reviewer may be thinking of those papers in contrasting OP50 (actually DA837) and HB101 (eg Shtonda and Avery, 2006, J Exp Biol).

We provide a shorter summary of this information in the revised paper (lines 383-390).

2. Multiple studies have investigated the density dependence and food choice in C. elegans. A recent paper studies spot leaving in a multiple patch arena of the same bacteria or different bacteria at different densities (Madirolas et al., 2023) and found that to first order the preference is entirely determined by bacterial density. This is the basis for a second potential confound in this study: The authors study multiple E. coli mutants, most of which act on bacterial metabolism, which very likely impacts growth rate and culture density, and possibly impacts cell size. At minimum, a control for cell density and CFU counts should be presented to see if density is a concern, beyond just OD600 before spotting on the plate. In addition, as bacteria were allowed to grow on NGM agar (a minimal medium, but nonetheless bacteria will grow), which could further exacerbate density differences. This confound could also explain the fact that glucose supplementation suppressed aversion to the crp mutant.

The reviewer makes a good point about bacterial density, which we address with **new data shown in Fig. 2e, Supplementary Figure 3 and the text (lines 189-196)**. Briefly, we directly measure bacterial counts of the three main *E. coli* strains, wild-type, Δcrp , and $\Delta cysE$, and vary the density of bacterial strains to ask whether this can explain their different effects on *C. elegans* behavior. We find that density is not a sufficient explanation for aversion, so a metabolic or dietary component is involved.

Supplementary Fig 3a shows the workflow for bacterial growth and supplementation in these assays. For density, we focus on cell number at the time of the assay, after growth on NGM agar (Supplementary Fig. 3d, e, j). We include OD600 for completeness (Supplementary Fig. 3b, c, i).

The Δcrp and $\Delta cysE$ mutants have moderate growth defects both in liquid medium and on NGM plates relative to wild-type *E. coli* (Supplementary Data 1 and Supplementary Fig. 3b-e). By varying the number of bacteria plated, we generated test plates that have different numbers of wild-type control, Δcrp , and $\Delta cysE$ bacterial cells (Supplementary Fig. 3b-e) and compared their effects in aversion assays (Supplementary Fig. 3f-h).

On wild-type *E. coli*, *C. elegans* responded to the bacterial food similarly across a ten-fold density range (Fig. 2e). Δcrp and $\Delta cysE$ were still aversive after controlling for bacterial cell number, but in interestingly different ways (Fig. 2e). For Δcrp , *C. elegans* exhibited consistent aversion at bacterial densities that were attractive wild-type *E. coli* (Fig. 2e, Supplementary Fig. 3e, g), suggesting that aversive metabolic or dietary factors contribute to Δcrp -induced aversion. For $\Delta cysE$, behavioral avoidance was rescued at the highest bacterial densities and was also rescued by acutely supplementing low bacterial densities with cysteine or its precursor *O*-acetylserine (Fig. 2e, Supplementary Fig. 3e, h). Importantly, this chemical supplementation did not necessarily increase bacterial density (Supplementary Fig. 3i, j, “post” entries). This result suggests a cysteine-related metabolic or nutritional deficit in $\Delta cysE$ beyond reduced cell number.

We agree that the Madirolas paper is a terrific contribution to the field, and now cite it (lines 189, 347). Madirolas *et al.* used a clever paradigm involving choices between multiple food patches to show that bacterial cell density is an initial preference “rule of thumb”. Compared to the assay used here, Madirolas *et al.* used a short-term 2-hour choice assay (we do not observe aversion behaviors until >4 hours on a lawn), seeded the bacteria at a lower density, and treated bacteria with antibiotics on a nutrient-poor substrate to minimize bacterial growth and metabolism; they specifically note in their paper that their assay is uncoupled from the effects of toxins and other metabolites. Our hypothesis is that our longer-term lawn aversion assay assesses metabolic effect of the food on *C. elegans*, and does not contradict a different “rule of thumb” for short term choice – indeed, we see that naïve animals will initially enter a lawn of even highly pathogenic bacteria, and only leave after several hours. Parenthetically, in comparisons to other *E. coli* strains, Madirolas *et al.* was the paper that found that *E. coli* B strains were more attractive than expected (comment 1).

3. Fluorescence reporters of stress pathways:

A potential concern for the induction of the stress genes is the observed developmental and feeding difference between the bacterial strains: It would be better to normalize to either animal size and/or a stable marker to ensure that the size is not a concern or a driver of the observed effects.

Yes, animal size can affect fluorescence quantification. In our assays we quantified mean fluorescence intensity per pixel rather than integrated intensity, as this approach is less sensitive to differences in animal size. However, to address the reviewer’s concern more completely, we have directly compared animal size using the exact bacterial strains, *C. elegans* reporter strains, and conditions as were used in Figure 2b and 2c. The sizes of each strain were comparable when fed on wild-type *E. coli*, Δcrp or $\Delta cysE$, although the *gst-4* reporter strain was slightly small in all conditions. We noted this in Methods (lines 581-583) and can include these results in a supplemental figure if desired.

4. The authors then move on to investigate which pathways in the worm are responsible for food aversion. They present data for the major neuromodulators. Unfortunately, only one allele is shown per neuromodulator, and at least for 5-HT the two alleles have quite some behavioral differences, specifically in the severity of the feeding and locomotion defects. It would be good to confirm the major results with a second allele.

This is a good point. In addition to the *tph-1(mg280)* serotonin mutant characterized throughout the paper (eg Fig. 3a), we have now examined *tph-1(n4622)* mutants on different *E. coli* diets and found similar behavioral defects as those of *tph-1(mg280)*. **These results have been added as Supplementary Fig. 4a and are mentioned in lines 208-210.**

Similarly, while we used the tyramine/octopamine *tdc-1(n3419)* allele in most figures (eg Figure 4d, 6d, 7e-f) we include *tdc-1(n3420)* in Figure 7g-i; both alleles demonstrating consistent behavioral responses across the three *E. coli* diets.

We have only examined a single *tbh-1* allele but have confirmed a role of the octopaminergic neuron RIC through rescue of the biosynthetic enzyme *tdc-1* and through tetanus toxin expression (Figure 4e-f).

5. The expression of CDNA of *ser-5* in pharyngeal neurons is unexpected - why did the authors assume *ser-5* to be relevant in pharyngeal neurons? Based on single-cell RNAseq, *ser-5* is only in a single pharyngeal neuron class (I3) that isn't implicated in being important for feeding. Instead, *ser-7* is the relevant 5-HT receptor in the pharynx (as seen in a later figure). How do the authors envision pharyngeal *ser-7* acts on a locomotion phenotype 'aversion'?

ser-5: We do not observe rescue in pharyngeal neurons. We typically scanned a number of receptor-expressing cells for each receptor mutant. According to the most recent version of CENGEN, *ser-5* is expressed at similar levels in RIM (where it rescues) as in pharyngeal I3

(where it does not rescue) and at lower levels in RIC (where it rescues) and pharyngeal I6 (where it does not rescue).

ser-7: Leon Avery and colleagues showed that M4 and MC pharyngeal neurons express *ser-7* to regulate pharyngeal pumping, but we did not observe rescue of aversion in either of these cells. We saw partial rescue in I1 pharyngeal neurons and reproduced the known effects of *tph-1* and *ser-7* on pharyngeal pumping but did not observe rescue of pumping in I1, so pumping and aversion may be different functions of *ser-7* (Supplementary Fig. 4b). We discuss these results in lines 276-280. Not much is known about I1 neurons; they stimulate pumping off food and can stimulate pumping when activated optogenetically (Trojanowski *et al.*, 2014, J Neurophysiol) and have a role in suppressing pumping in response to aversive light (Bhatla *et al.*, 2015, Current Biology). With respect to regulation of locomotion, I1 is unique in pharyngeal neurons in having gap junctions to the extrapharyngeal nervous system. It could affect locomotion through those connections or through long-range release of neuropeptides, but we do not have specific information at this time, so we did not speculate about these points.

6. The feeding rate measurements on the selected strains are a bit surprising: Previous studies suggested that pumping is reduced on 'mediocre' foods (Shtonda and Avery, 2006). The size difference in bacteria is interesting, and not unexpected given the specific pathways affected. However, it also adds another confound as it has been shown that ingestion is size-selective and it doesn't seem to match with the pumping rate data, which would be expected to be lower for larger bacteria. Taken together with the complex confound of satiety and density, I am unsure if the results are easily interpretable as stated.

We repeated the feeding experiments at higher resolution and confirmed that *C. elegans* has slightly reduced pumping rates on the *E. coli* mutants Δcrp and $\Delta wxxE$, which have a larger bacterial cell size (Supplementary Fig. 2a, feeding rate). This agrees with the observations of Shtonda and Avery (2003, 2006), but they studied L1 larvae, which have smaller pharynxes, whose feeding rates are more sensitive to bacterial size than adults. We note that Shtonda and Avery used the DA837 bacterial strain, which is particularly hard for young larvae to eat and is flagged as a poor-quality food for growth and feeding in the *Caenorhabditis* strain collection (<https://cgc.umn.edu/strain/DA837>). Bacteria that are hard to eat elicit stress responses and *daf-7* expression in ASJ, as shown by Boor, Meisel and Kim (2024, eLife), and this matches our results with Δcrp and $\Delta wxxE$.

There is a slightly elevated pumping rate in animals consuming $\Delta cysE$, so feeding and aversion responses can be separated.

7. Overall, the results section would be improved by providing a rationale for certain experiments e.g., specific rescues beyond stating significance or non-significance. I would therefore also recommend overall editing and stream-lining the presentation to clarify the specific findings and to contextualize these within the literature.

We appreciate these suggestions and have made a number of changes.

We have focused more on primary conclusions. For example, we highlighted key findings from Fig. 2 and Supplementary Fig. 2 and 3 by reorganizing the figures and revising text (lines 159-196).

We added justification for the choice of cells for receptor rescue experiments (lines 246-247, 276-280, 290-291, 317-318, 322-323). We also provide a caveat that the cells we identified may not be “unique” in allowing rescue – there may be other combinations that also rescue. We recently discovered that a neuropeptide receptor could fully rescue a mutant phenotype in several non-overlapping sets of cells.

These modifications provide clearer experimental justification while making the manuscript more accessible to a broad readership.

Minor concerns:

8. The description of the bacterial pathways in the results section is very lengthy and could be edited to be more concise while retaining the relevant information.

We have condensed the description of bacterial metabolism and eliminated some repetition in the revised text (lines 118-148).

9. Allele names are omitted in the main text. It would be nice to include these in the manuscript for easier comparability to the many other studies that have also used the main neuromodulator mutants.

We have been told by non-*C. elegans* colleagues that too many allele numbers can be distracting when included in the text. To compromise between completeness and clarity, we now include a full supplementary strain list (Supplementary Data 3) with each relevant figure panel noted.

We thank the reviewers for their support and intellectual contributions, which have improved the revised manuscript.